# Multiscale cardiac imaging spanning the whole heart and its internal cellular architecture in a small animal model

Graham Rykiel[1], Claudia S López[1,2], Jessica L Riesterer[1,2], Ian Fries[1], Sanika Deosthali[1], Katherine Courchaine[1], Alina Maloyan[3], Kent Thornburg[3], Sandra Rugonyi[1,3]*

[1]Biomedical Engineering, Oregon Health & Science University, Portland, United States; [2]Multiscale Microscopy Core, Oregon Health & Science University, Portland, United States; [3]Center for Developmental Health, Knight Cardiovascular Institute, Oregon Health & Science University, Portland, United States

**Abstract** Cardiac pumping depends on the morphological structure of the heart, but also on its subcellular (ultrastructural) architecture, which enables cardiac contraction. In cases of congenital heart defects, localized ultrastructural disruptions that increase the risk of heart failure are only starting to be discovered. This is in part due to a lack of technologies that can image the three-dimensional (3D) heart structure, to assess malformations; and its ultrastructure, to assess organelle disruptions. We present here a multiscale, correlative imaging procedure that achieves high-resolution images of the whole heart, using 3D micro-computed tomography (micro-CT); and its ultrastructure, using 3D scanning electron microscopy (SEM). In a small animal model (chicken embryo), we achieved uniform fixation and staining of the whole heart, without losing ultrastructural preservation on the same sample, enabling correlative multiscale imaging. Our approach enables multiscale studies in models of congenital heart disease and beyond.

*For correspondence:
rugonyis@ohsu.edu

Competing interests: The authors declare that no competing interests exist.

## Introduction

Congenital heart disease (CHD), which manifests as a morphologically defective heart, affects about 1% of newborn babies, and remains the primary cause of non-infectious children mortality in the developed world (*Gilboa et al., 2010*; *van der Linde et al., 2011*). While CHD mortality rates have been dramatically reduced in recent years thanks to advances in surgical practice and interventional technologies (*Gilboa et al., 2010*; *Czosek and Anderson, 2016*), CHD patients continue to be at an increased risk of developing heart failure at a much younger age than the general population (*Gilljam et al., 2019*). Despite early indicators of success, heart failure continues to take the lives of young children with CHD: 10% to 25% of newborns with a critical heart defect do not survive the first year, and 44% do not survive to 18 years of age (*Oster et al., 2013*; *Stout et al., 2016*). This unfortunate trend points to cardiac deficiencies in CHD that are not yet understood (*Sugimoto et al., 2015*).

Imaging approaches have been employed to study cardiac function in normal and CHD hearts (*Prakash et al., 2010*). In humans, ultrasound-based echocardiography, which can image the in vivo motion of cardiac walls and measure blood flow velocities within the heart, is used to diagnose fetal CHD in utero and assess CHD severity (*Arya et al., 2014*; *Jatavan et al., 2016*). After the baby is born, echocardiography is useful to monitor cardiac function before and after interventions to repair CHD. Magnetic resonance imaging (MRI) is also employed to precisely diagnose malformations and monitor cardiac function in CHD (*Sreedhar et al., 2005*; *Ntsinjana et al., 2011*). In heart

**eLife digest** The heart is our hardest-working organ and beats around 100,000 times a day, pumping blood through a vast system of vessels to all areas of the body. Specialized heart cells make the heart contract rhythmically, enabling it to work efficiently. Contractile molecules inside these cells, called myofibrils, align within the heart cells, and heart cells align to each other, so that the heart tissue contracts effectively.

However, when the heart has defects or is diseased this organization can be lost, and the heart may no longer pump blood efficiently, leading to sometimes life-threatening complications. For example, around one in a hundred newborn babies suffer from congenital heart defects, and despite medical advances, these conditions remain the main cause of non-infectious mortality in children.

Many cases of congenital heart disease are diagnosed before a baby is born during an ultrasound scan. However, these scans, as well as subsequent diagnostic tools, lack the precision to detect problems within the heart cells. Now, Rykiel et al. used two complementary imaging techniques known as micro-computed tomography and scanning electron microscopy to acquire pictures of the whole heart as well as of the organization inside the heart cells.

This made it possible to capture the structure of the heart tissue at both micrometer (the whole heart) and nanometer resolution (the inside of the cells), and to study what happens within the heart and its cells when the heart has a defect.

Rykiel et al. tested the imaging technology on the hearts of chicken embryos, at stages equivalent to a five to six-month-old human fetus, and compared a healthy heart with a heart with a defect called tetralogy of Fallot. They found that the tissues in the heart with a defect had a sponge-like appearance, with increased space in between cells. Moreover, the myofibrils of the heart with a defect were aligned differently compared to those in the normal heart.

More research is needed to fully understand what happens when the heart has a defect. However, the imaging technology used in this study offers the possibility of examining the heart at an unprecedented level of detail. This will deepen our understanding of how structural heart defects arise and how they affect the pumping of the heart, and will give us clues to design better treatments for patients with heart defects and other heart anomalies.

development animal research, moreover, optical coherence tomography (OCT) and echocardiography are used with avian and mammalian models of CHD. OCT, like ultrasound, is a non-invasive technique that can image heart motion and measure blood flow velocities within the heart. OCT resolution (<5 μm) is ideal for early avian and mouse embryos during tubular-heart developmental stages (*Larina et al., 2009*; *Syed et al., 2011*; *Larina et al., 2011*; *Peterson et al., 2017*; *Jenkins et al., 2007*). For later stages of heart development echocardiography is used in mice and chick (*Grune et al., 2018*; *Damen et al., 2017*; *Midgett et al., 2017a*). In small animal research, moreover, due to the small size of developing hearts, functional imaging techniques are frequently complemented with micro-CT or histology to more accurately phenotype the heart (*Midgett et al., 2017a*; *Butcher et al., 2007*; *Gregg and Butcher, 2012*).

The structural (morphological or 'geometrical') characteristics of heart malformations, including changes in cardiac wall architecture, have been extensively studied. However, the mechanisms that lead to an anomalous cardiac architecture in CHD and the clinical consequences of it are unknown. Recent studies have revealed an abnormal orientation of myocardial cells (the heart muscle cells) within CHD hearts (*Garcia-Canadilla et al., 2019*; *Stephenson et al., 2018*; *Garcia-Canadilla et al., 2018*). Myocardial cells are elongated, cylindrical-like cells, that contract along their long axis. In a normal heart, myocardial cells arrange in sheet-like layers with their long axes in parallel to each other, forming a helical pattern (*Gilbert et al., 2007*; *Omann et al., 2019*). This highly organized pattern ensures that cardiac contraction occurs along specific directions, so that the heart can effectively eject blood into the pulmonary and systemic circulations. Perturbations of the normal myocardial architecture affect cardiac contractility and compliance. This is both because the tissue can no longer contract on the very specific directions and patterns that optimize cardiac function, but also because cardiac contraction force may be diminished. Therefore, changes in myocardial architecture

(with respect to normal) frequently lead to heart tissues with impaired contractility and reduced compliance. Newly developed contrast episcopic microscopy and synchro micro-CT imaging techniques, enable non-destructive analyses of banked human fetal and neonatal hearts with CHD (*Stephenson et al., 2018*; *Garcia-Canadilla et al., 2018*). These emergent studies are revealing myocardial disarray in CHD (with respect to their normal counterparts) that very likely affect cardiac function before and after surgical repair. In addition to changes in the myocardial organization, the subcellular contraction machinery of myocardial cells (e.g. the myofibrils that contract the cell; and the mitochondria that provide energy for contraction) may also be compromised in CHD, affecting heart function. For instance, a reduced number of myofibrils per myocardial cell will reduce the contractile force that the cell can generate, leading to cardiac dysfunction. The extent to which the cells of malformed hearts exhibit deficiencies is unknown (*Garcia-Canadilla et al., 2019*; *Garcia-Canadilla et al., 2018*; *Sanchez-Quintana et al., 1999*). This is in part due to limitations of existing technologies that have not achieved precise multiscale mapping to decipher the association between structural and cellular deficiencies in the heart and beyond.

We describe here a proof-of-concept correlative multiscale procedure that combines imaging of whole heart morphology and its subcellular organization (ultrastructural organelle architecture). Our multiscale procedure uses micro computed tomography (micro-CT) imaging to capture heart morphology at micrometer resolution, and scanning electron microscopy (SEM) to capture cardiac tissue ultrastructure at nanometer resolution. Current SEM technologies allow for three-dimensional (3D) imaging, enabling reconstruction and quantification of ultrastructural features within a tissue volume (*Rennie et al., 2014*; *Midgett et al., 2017b*; *Hussain et al., 2018*). Among 3D SEM methods, we have selected serial block-face SEM (SBF-SEM) for ultrastructural imaging, as it allows 3D imaging of relatively large volumes (sample size $40 \times 60 \times 40$ $\mu m^3$). The methodology we present herein improves upon previous protocols by achieving uniform staining of a relatively large heart sample (4-5 mm wide), circumventing micro-CT X-ray penetration issues, and allowing sample screening and selection prior to full SEM sample preparation. Our multiscale imaging, further, enables mapping of structural and ultrastructural heart features.

As proof of concept, we applied our developed multiscale imaging procedure to two embryonic chick hearts. These hearts were collected at stages corresponding to about 5–6 months of human fetal development, when the heart is already formed but maturing in preparation for birth/hatching. We imaged: (1) a control heart with no structural defects and (2) a heart with tetralogy of Fallot (TOF), a combination of structural heart malformations found in humans (*Bailliard and Anderson, 2009*; *Wiputra et al., 2018*). Our results show the morphology and ultrastructure of these two hearts, and emphasize the need for a multiscale approach to deepen our understanding of CHD and enable the development of effective strategies to combat heart failure in CHD.

## Results

### Overview of multiscale imaging procedure

To achieve multiscale imaging, we followed a four-step protocol (see *Figure 1*; details in Materials and methods). Briefly, in *Step 1* the heart was excised, homogeneously fixed and stained for micro-CT. Initial staining followed a modified ferrocyanide-reduced osmium–thiocarbohydrazide–osmium (ROTO) protocol (*Hua et al., 2015*; *Malick and Wilson, 1975*; *Willingham and Rutherford, 1984*) typically used for electron microscopy (EM) sample preparation. Three-dimensional micro-CT images (10 μm resolution) confirmed uniform ROTO staining of the whole heart and provided morphological cardiac details. At this time, the heart samples were stored until further processing, enabling selection of specific samples for full processing based on micro-CT scans. *Step 2* finalized the preparation of the whole heart for SBF-SEM by post-staining the hearts with uranyl acetate and lead aspartate and then embedding them in a resin block. Uniform staining was confirmed on a semithin (250 nm) section of the block, which also determined ultrastructural quality and enabled registration to micro-CT images. In *Step 3*, a slab of the sample was cut and sectioned around a specific region of interest (ROI, ~$500 \times 500$ $\mu m^2$) from which sub-ROIs for 3D SBF-SEM imaging (~$40 \times 60$ $\mu m^2$) were selected. In *Step 4*, 3D SBF-SEM datasets were acquired (10 nm lateral resolution and 40 nm depth resolution).

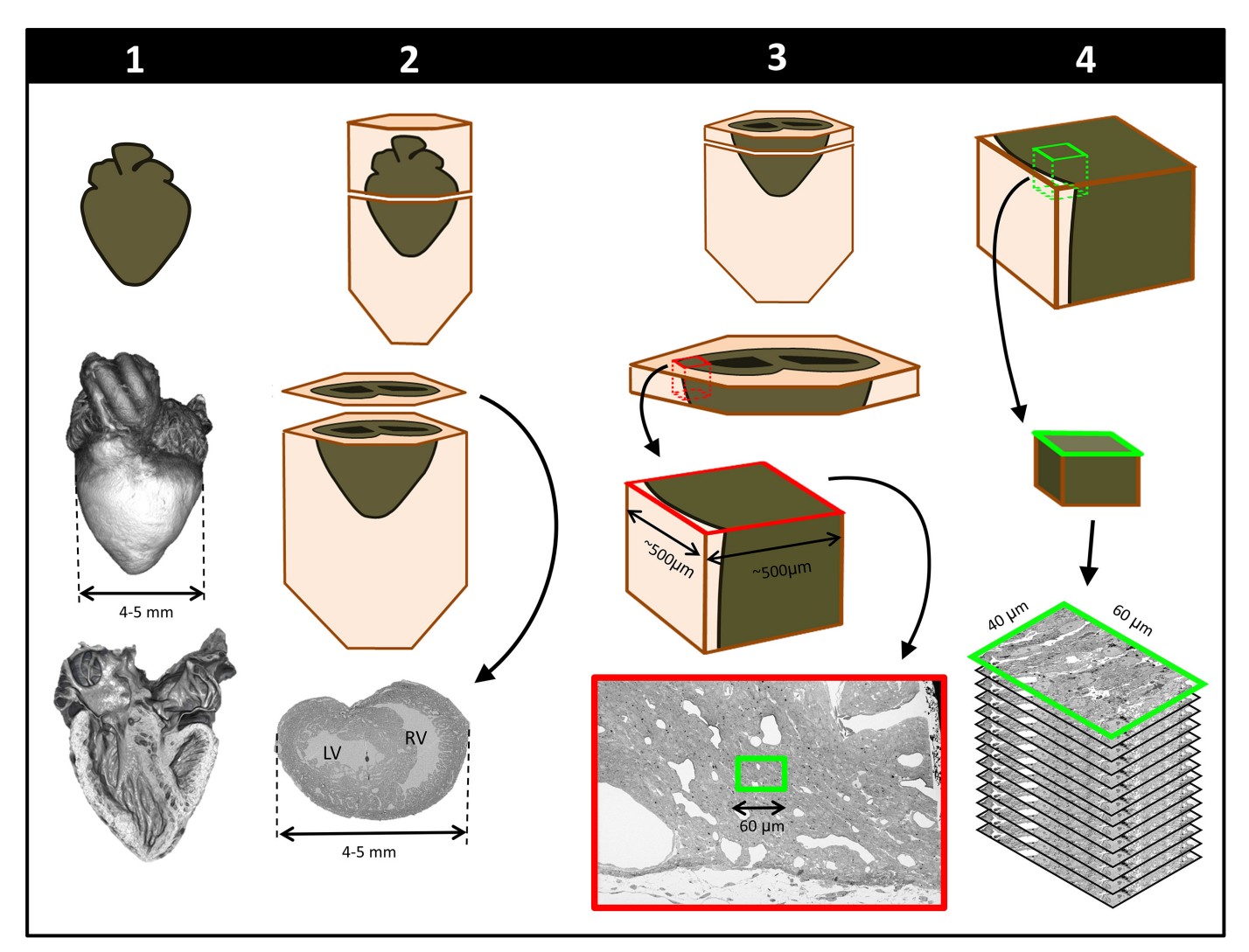

**Figure 1.** Schematics of steps performed to achieve cardiac multiscale imaging, which yields both 3D whole-heart images and 3D ultrastructural images from the same heart. Columns depict the four steps employed. In *Step 1*, the heart is post-fixed with osmium tetroxide to provide contrast for 3D micro-CT images of the whole heart (middle row). Digital sections of the micro-CT scans (bottom row) reveal the heart's interior and allow for cardiac phenotyping and assessment of stain penetration. In *Step 2*, contrast staining is finalized and the resin block in which the heart is embedded is sectioned to reach a plane of interest (bottom row). In *Step 3*, after cutting a slab of the sample, a region of interest (ROI) is sectioned from the slab, mounted, and then scanned by SEM backscattered imaging methods to aid in the selection of sub-ROIs (for example, the sub-ROI highlighted in green). In *Step 4*, the selected sub-ROI 3D SBF-SEM images are acquired by progressively sectioning and imaging layers that are 40-nm apart.

## Whole heart imaging

We obtained 3D micro-CT images of the whole heart, featuring both external and internal structures at 20 µm resolution (*Figure 2*; *Step 1* in *Figure 1*). Despite the relatively large dimensions of the heart (5-6 mm long), tissue contrast was uniform across the heart walls and septae and allowed us to visualize the microstructural details of the heart chambers, valves, and great arteries.

## Cardiac structure analysis from 3D micro-CT images

Micro-CT images guided selection of two hearts for subsequent SBF-SEM imaging and analysis. We selected: (1) a normal heart and (2) a heart exhibiting TOF malformation. In a normal heart, blood in the left ventricle (LV) and right ventricle (RV) is separated by an interventricular septum; the pulmonary valve and pulmonary artery connect to the RV, which pumps blood to the lungs; and the aortic valve and aorta connect to the LV, which pumps blood to the body. TOF is characterized by a

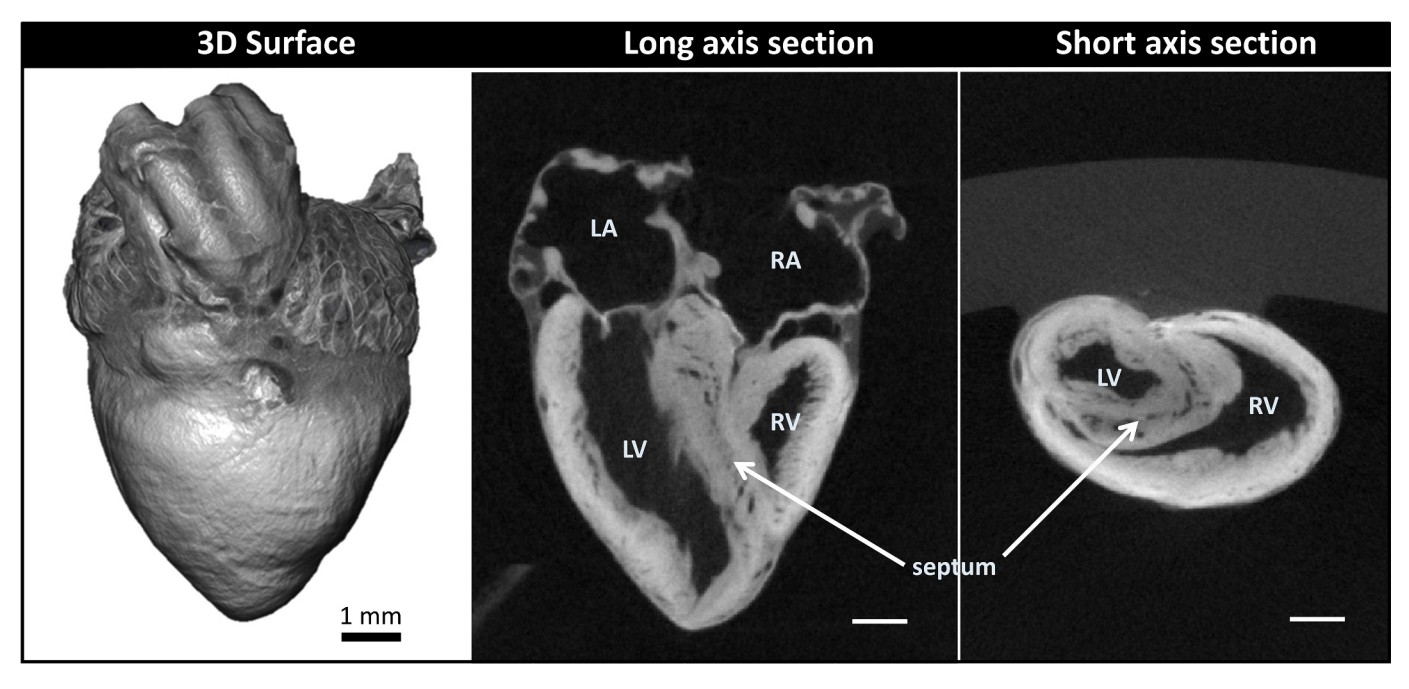

**Figure 2.** Micro-computed tomography (micro-CT) images of a chicken embryo normal heart. The contrast was achieved by following *Step 1* of our correlative multiscale imaging protocol (*Figure 1*). *From left to right:* External 3D surface of the heart; cardiac section along the heart's long axis (coronal section); cardiac section across the heart's short axis (transverse section). Cardiac sections show uniform staining across cardiac walls, and reveal the heart internal and external microstructure. LA: left atrium; LV: left ventricle; RA: right atrium; RV: right ventricle. Scale bars 1 mm.

combination of four defects: (i) ventricular septal defect, which is a hole in the interventricular septum; (ii) overriding aorta, a change in the position of the aorta such that it sits in the middle of the two ventricles, on top of the ventricular septal defect; (iii) pulmonary stenosis or atresia, a narrowing or closure of the pulmonary artery or pulmonary valve; and (iv) RV hypertrophy, a thickening and enlargement of the RV wall. RV hypertrophy in TOF, however, develops over time as the stenosis of the pulmonary artery increases pressure in the RV after birth (*Bailliard and Anderson, 2009*) and was not present in the heart examined in this study (see *Figure 3* for a comparison of the selected normal and TOF hearts). The TOF heart analyzed here featured supravalvular pulmonary stenosis, a ventricular septal defect, and an overriding aorta. The right ventricle was enlarged compared to the control heart (*Figure 3*). Further, the TOF heart was missing the left branch of the pulmonary artery. In humans, this rare condition, called unilateral absence of a pulmonary artery, is known to occur in conjunction with TOF or cardiac septal defects (*Reading and Oza, 2012*).

From micro-CT images of the two hearts, we quantified cardiac structural features for illustration purposes. It is important to mention, however, that these quantifications only pertain to the two hearts compared in this study, and thus are not necessarily representative of normal nor TOF heart populations. For the two hearts compared, we found that the LV volume was about the same in the normal, control (CON) heart and TOF hearts. However, in the CON heart the RV volume was 40% smaller than the LV volume (RV volume/LV volume ~0.6), while in the TOF heart RV volume was 30% larger than the LV volume (RV volume/LV volume ~1.3). Overall the volume of the TOF RV was about twice as big as the CON RV, suggesting functional impairment of the TOF RV.

## Cardiac ultrastructural imaging

We chose to characterize the *ultrastructural* architecture of the selected hearts at approximately the transverse section at which the heart width (from LV to RV wall) is maximal, also referred to as the equatorial plane. Images of semithin cross-sections for each heart (*Figure 4*, top row; corresponding to *Step 2* in *Figure 1*) show that staining was uniform, indicating successful stain penetration through

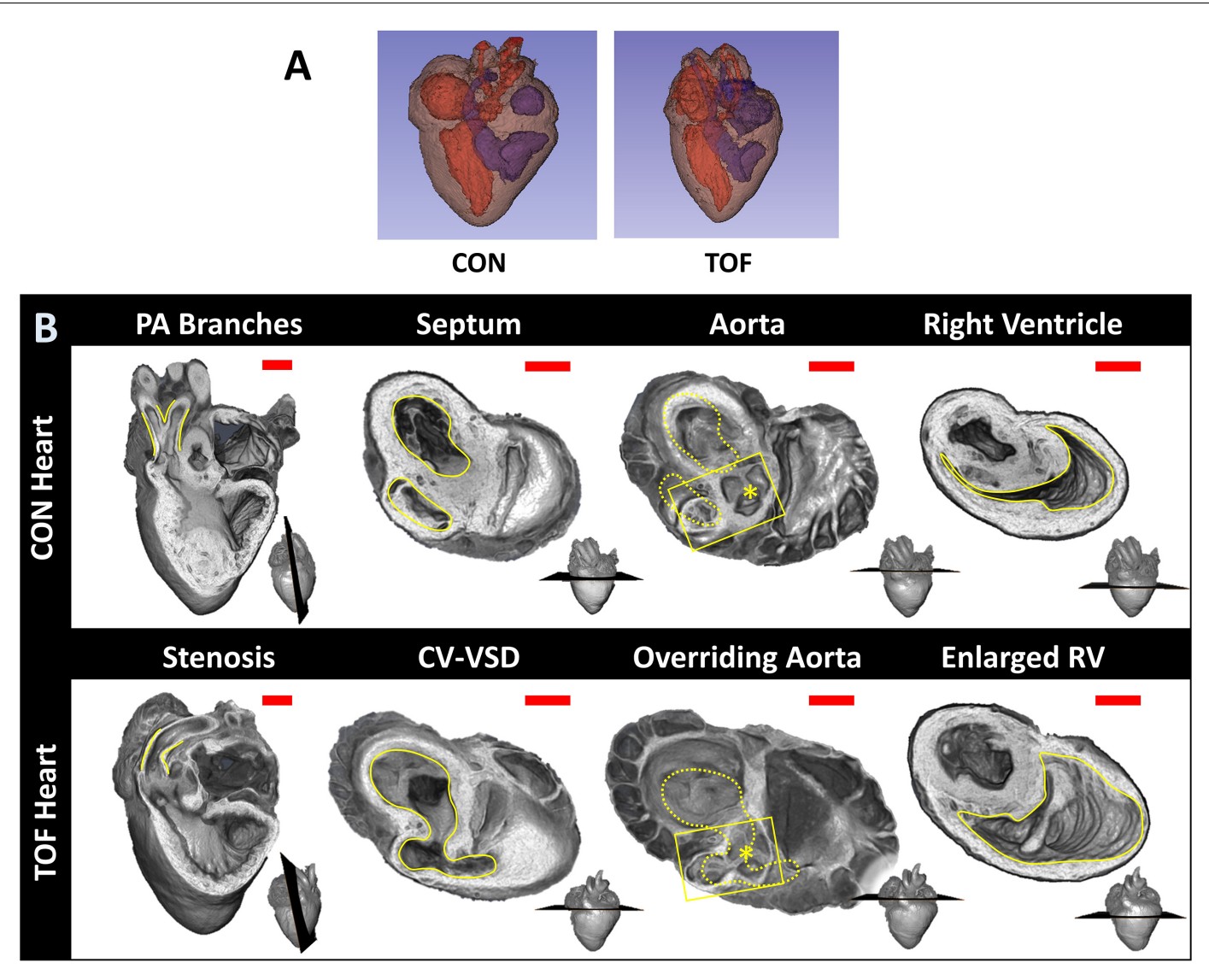

**Figure 3.** Comparison of micro-CT images of the two hearts selected for this study. (**A**) Segmentations showing the heart morphology for the normal, control (CON) heart and the TOF heart. Red: lumen of the left atrium and ventricle as well as aorta; Blue: lumen of the right atrium and ventricle as well as pulmonary artery; Brown: heart tissue. (**B**) Detailed comparison of the two hearts. Each column compares a cardiac feature (highlighted in yellow) between the hearts. The position of the plane along which the tissue was cut for display is shown at the bottom right of each image. From left to right: *Pulmonary artery (PA) branches:* On the control heart, the PA is bifurcated (yellow lines) whereas the left branch of the PA is absent in the TOF heart. The remaining PA of the TOF heart exhibits supravalvular stenosis (yellow lines). *Septum:* The ventricles (yellow lines) in the control heart are discrete, separated by an intact interventricular septum, whereas the TOF heart shows a conoventricular septal defect (CV-VSD). *Aorta:* In the control heart the aortic valve (asterisk) is connected to the left ventricle, whereas in the TOF heart the aortic valve is positioned directly over the VSD. Dotted yellow lines show the position of yellow lines in the septum column. *Right Ventricle (RV):* The RV (yellow lines) is significantly larger in the TOF heart than in the control heart. Scale bars = 1 mm.

the heart tissues. Please note that the transverse section of the TOF heart was below the ventricular septal defect and thus exhibits a continuous septum.

SEM images of semithin sections guided selection of ROIs from each heart. For this study, we selected two ROIs within each heart LV, denoted by ROI A and ROI B. After cutting the sample, the ROIs were first imaged with SBF-SEM at low resolution (65–80 nm lateral resolution; *Step 3* in *Figure 1*), from which sub-ROIs were further selected (see *Figure 4*). These sub-ROIs (~40×60 μm²) were imaged at 10 nm lateral resolution, and we acquired 800–1000 images in depth (with ~40 nm of depth distance, thus 32 to 40 μm in depth). These high-resolution images showed the

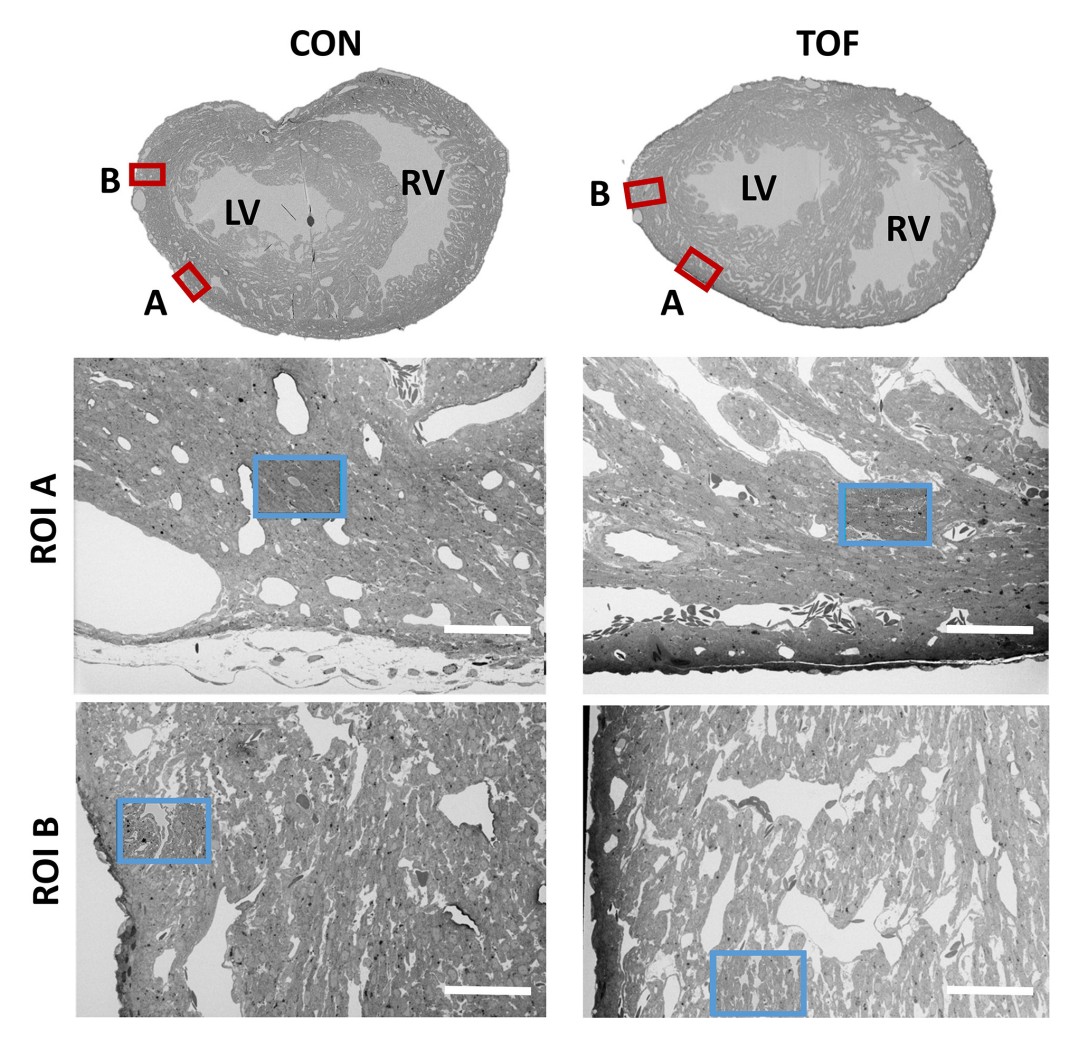

**Figure 4.** SBF-SEM images of the control (CON, left) and TOF (right) hearts, indicating the location of imaging. Overview scans from semithin transverse sections of each heart (*top row*) were used to confirm uniform stain penetration and the location of 2 ROIs per heart (ROI A and ROI B) further analyzed (red boxes). Semithin sections are not to scale. Blood vessels and trabeculae served as landmarks for accurate positioning of the ROIs and sub-ROIs (blue boxes). LV: left ventricle; RV: right ventricle. The sub-ROIs corresponding to ROI A (*second row*), were fully segmented (nuclei, extracellular space, myofibrils, and mitochondria). However, the sub-ROIs corresponding to ROI B (*third row*) were only partially segmented (only a fraction of the images in the stack were segmented) for quantification purposes. Scale bars 60 μm.

conservation of ultrastructural features (see *Figure 5*). Images exhibited continuous nuclear membranes, intact mitochondria, and defined myofibrils, indicating that we achieved both appropriate and uniform fixation and staining of the hearts with our protocol.

## 3D SBF-SEM image reconstruction

SBF-SEM image stacks provided 3D volumetric reconstructions of sub-ROIs. While volumetric image resolution was not isotropic (10 nm lateral resolution versus 40 nm depth resolution), ultrastructural features could be visualized from any angle of view within the reconstructed images (see *Figure 6*). Thus SBF-SEM images allowed us to visualize the orientation and organization of nuclei, myofibrils, and mitochondria (among other features) in heart tissue samples.

## SBF-SEM image segmentation and quantification

To more easily visualize and quantify cardiac ultrastructure, we segmented (delineated) from SBF-SEM images the cell nuclei, myofibrils, mitochondria, and the extracellular space. To this end, we

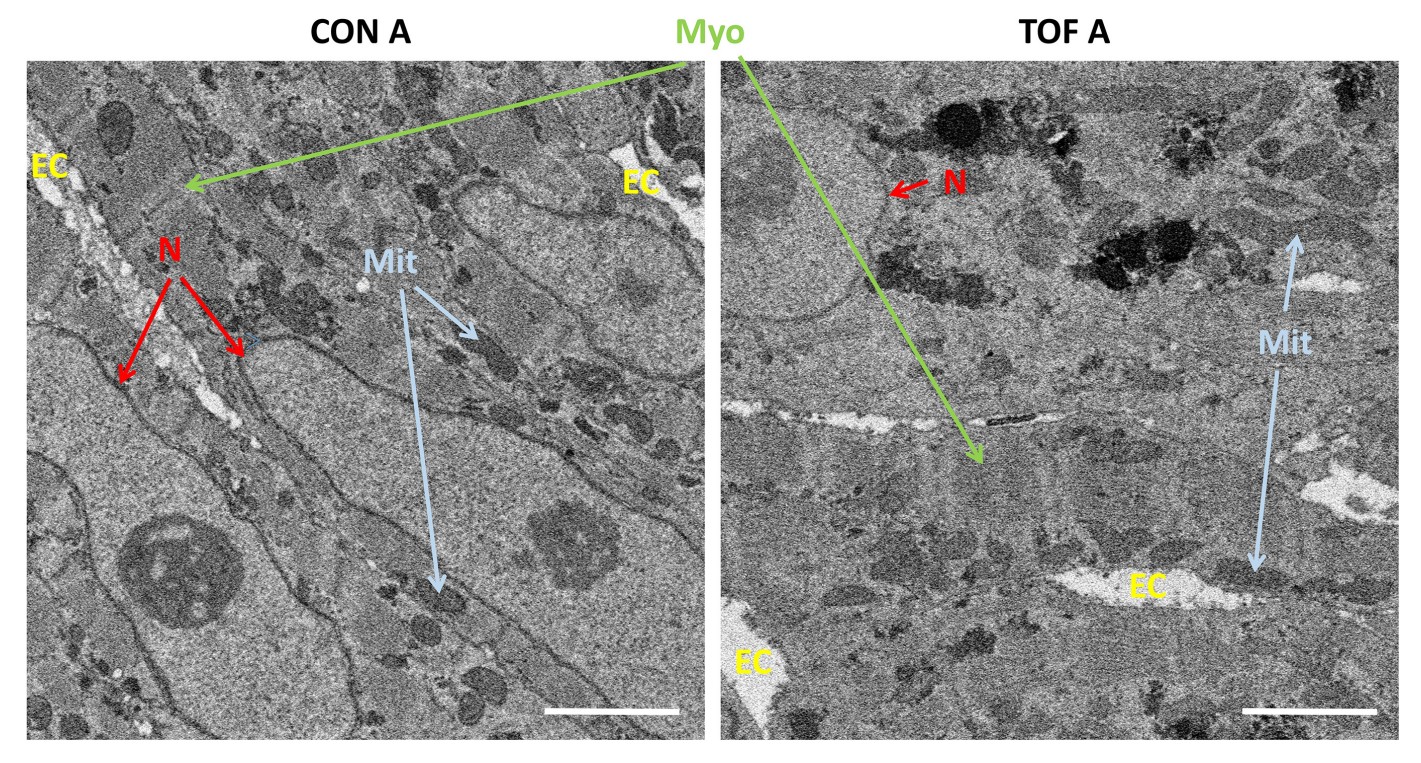

**Figure 5.** Detail of high-resolution SBF-SEM images obtained. The pictures depict small regions within the selected sub-ROIs from region A of the control (CON) and TOF hearts. Nuclear membranes (N) are intact, as well as myofibrils (Myo) and mitochondria (Mit). Finally, the extracellular space (EC) is also visible. Scale bars 2 μm.

used a combination of deep learning algorithms and tools available on the Dragonfly 4.1 software (Object Research Systems, Quebec, Canada). When independently tested against carefully annotated images (two each from the CON and TOF hearts, region A), the segmentation accuracy from the deep learning algorithm was at least 90% for myofibrils, 94% for mitochondria, and 98% for nuclei. Additional manual segmentation 'clean up' was thus required to improve the accuracy of organelle depictions. These segmentations, however, reveal the detailed 3D ultrastructural architecture of the heart wall (e.g. see *Figure 7*).

Ultrastructural quantifications of the two hearts were performed for illustration purposes. To control the accuracy of segmentations for quantification, we used subsets of the full SBF-SEM datasets by selecting images and regions within images from the complete dataset. From these selected image regions, we quantified the percentage of the cell occupied by nuclei, myofibrils, and mitochondria. That is, we quantified selected organelle density within cells. We found that, for our two hearts, the density of nuclei, myofibrils and mitochondria was similar between the CON and TOF hearts (see *Figure 8A*). We also quantified the percentage of the image regions occupied by extracellular space. The TOF heart exhibited more extracellular space than the CON heart (see *Figure 8B*), which was also consistent with a visual inspection of segmented extracellular space images (see *Figure 8C*).

Visualization of segmentations of the entire sub-ROIs from region A of the CON and TOF hearts revealed a slightly different orientation of myocardial cells between samples (*Figure 9*). Quantification of myofibril elliptical and transmural angle orientations (see *Figure 10A*) revealed that in the CON heart the myofibrils in the left ventricle were oriented at around 45° in the radial direction with respect to the circumferential directions (θ angle or transmural angle) with a standard deviation of approximately 32°. The myofibril elliptical angle, defined here as the angle between the circumferential and longitudinal directions (Φ angle) was around 70°, and exhibited a wide dispersion, with a standard deviation of 36° (*Figure 10B*). In contrast, the myofibrils in the TOF heart were oriented

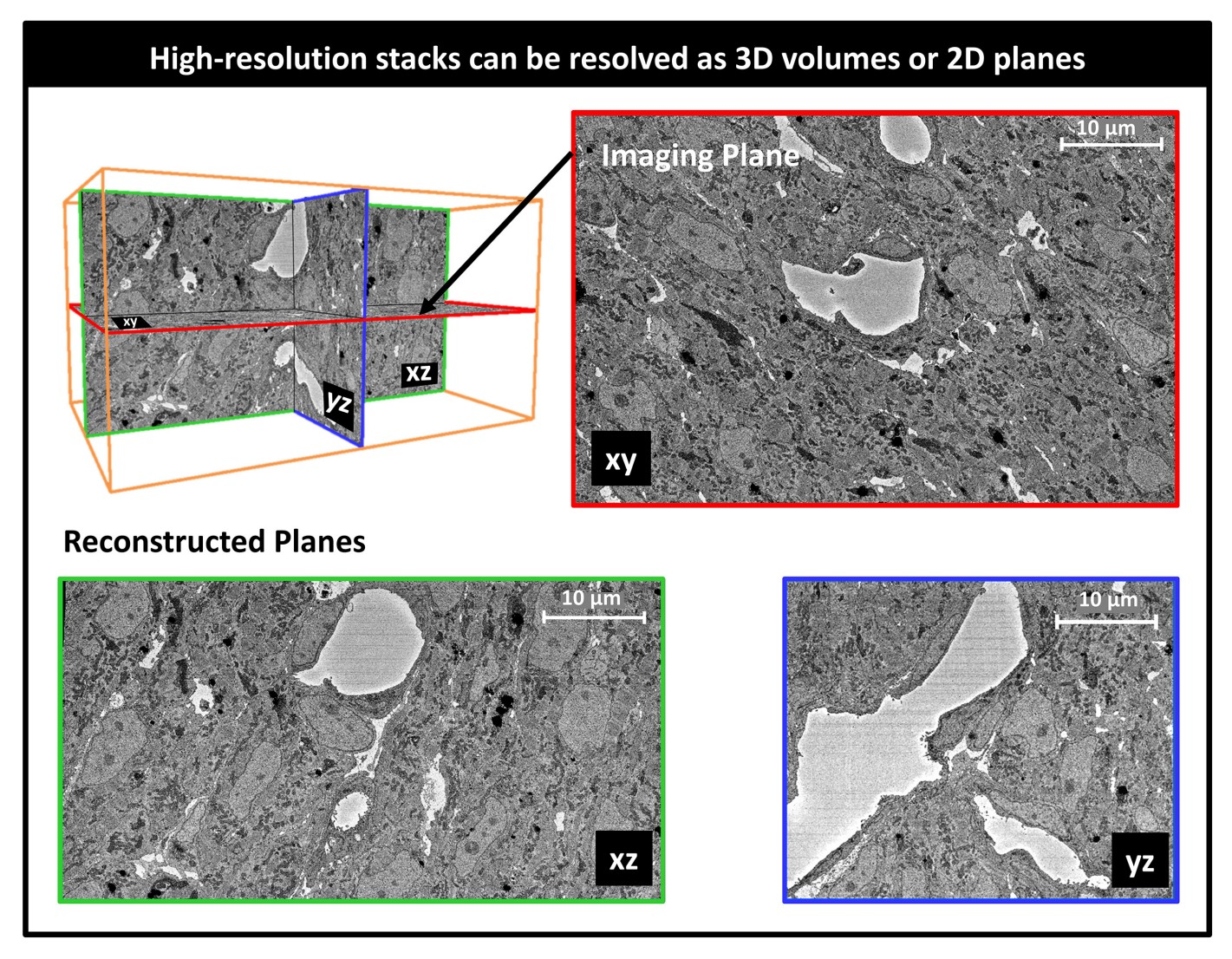

**Figure 6.** Example 3D reconstruction of SBF-SEM image stack acquired from the control heart. xy is the imaging plane, acquired at 10 nm lateral resolution. z is the depth direction, with xy images acquired every 40 nm. xz and yz are reconstructed perpendicular planes that show the continuity of the ultrastructural features along the z-axis (depth), and thus alignment of the acquired images. Scale bars = 10 μm.

almost circumferentially in the transmural plane, with an average transmural θ angle of 7° (20° standard deviation); while the elliptical angle Φ was around 15° with a standard deviation of 15°. Thus in the TOF heart, at the sample location, myofibrils run closer to the circumferential direction than in the CON heart (see *Figure 10C*). It is important to notice that these quantifications and differences pertain only to the two hearts under consideration (and samples within the heart) and do not reflect population trends. A more detailed study including more animals is needed to infer population differences.

## Discussion

Heart function relies on a multiscale, finely orchestrated contractile machinery. Efficient heart pumping requires a highly organized cardiac architecture linked to the cell metabolism (*Spotnitz et al., 1966*; *Feuvray, 1983*; *Sahli Costabal et al., 2019*). Structural malformations in the chambers, valves or vessels of the heart, together with disruptions to the organization or number of cardiac cells, and/ or their ultrastructure, can compromise the heart's ability to pump blood efficiently (*Garcia-*

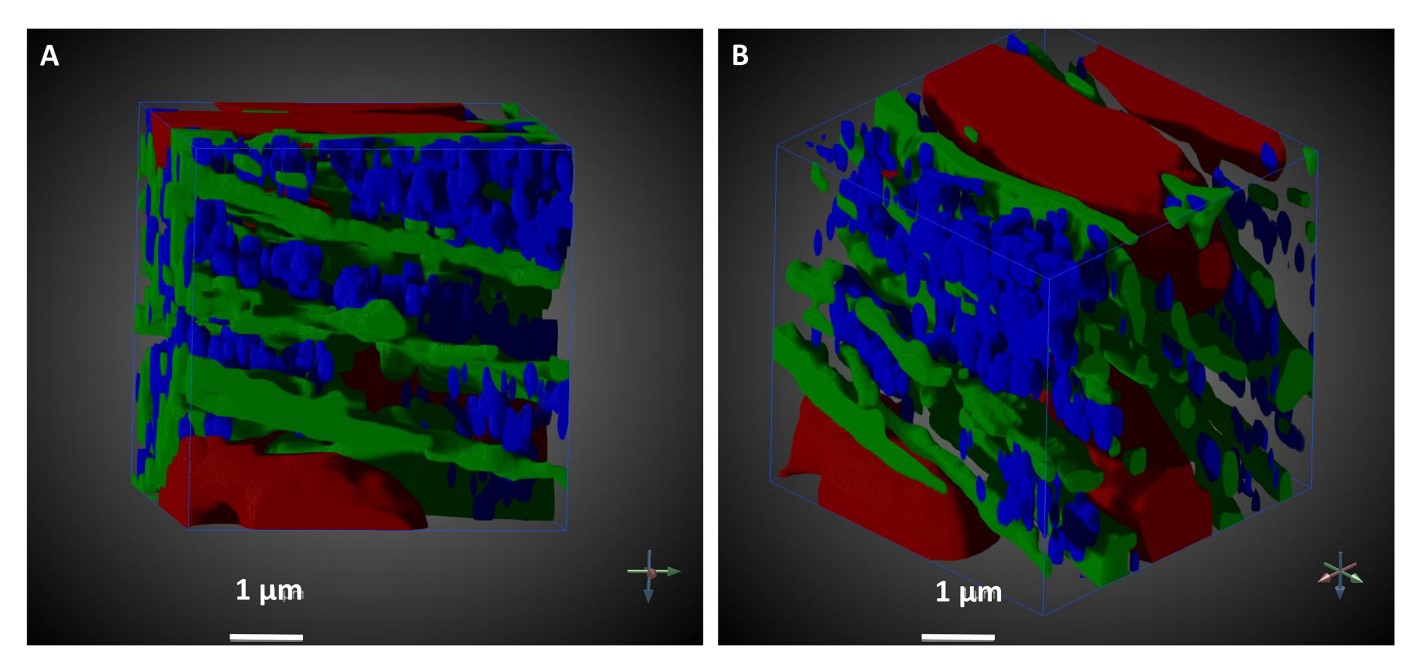

**Figure 7.** Segmented portion of 3D SBF-SEM heart tissue images. The pictures illustrate the level of detailed ultrastructural architecture that can be obtained from 3D SBF-SEM images. (**A and B**) show the same $10 \times 10 \times 10 \ \mu m^3$ of the LV developing chick heart tissue, TOF heart, from different points of view. Arrows indicate coordinate system, and its rotation. Organelle segmentation is color-coded. Red: nuclei; Blue: mitochondria; Green: myofibrils. Scale bars = 1 μm.

*Canadilla et al., 2019*; *Garcia-Canadilla et al., 2018*; *Sanchez-Quintana et al., 1999*; *Pinali et al., 2015*; *Holzem et al., 2016*). Anomalous microstructural and ultrastructural architectures can be detrimental to the heart's capacity to adapt to new conditions imposed by corrective surgeries or other therapies intended to repair structural congenital heart defects. Thus, it is to our advantage to understand the organization of the heart at multiple scales, and how it is disrupted in CHD, to advance treatment planning.

The heart walls contain myocardial cells, which are elongated, cylindrical-like muscle cells that are aligned in patterns that optimize cardiac contractility (*Fischman, 1967*; *Brook et al., 1983*). In humans, myocardial cells are about 50–150 μm long and 10–20 μm thick (diameter), and in the heart are connected to each other forming a 3D network (*Gilbert et al., 2007*). More specifically, myocardial cells are organized in laminar sheets that exhibit characteristic cell orientations (elliptical and transmural angles), which change over the heart wall thickness and the cardiac cycle (*Gilbert et al., 2007*; *Sonnenblick et al., 1967*). At the ultrastructural level, myocardial cells contain contractile units, the myofibrils, which are supplied with energy (ATP) by the mitochondria surrounding them. In a healthy, mature heart, myocardial nuclei have an ellipsoidal shape aligned with the cell long axis; myofibrils are meticulously aligned and organized along the myocardial cells; and mitochondria are densely packed around the myofibrils (*Hussain et al., 2018*; *Brook et al., 1983*). Other cell components (such as lipid droplets and glycosomes) are organized around the cell myofibrils, mitochondria and nuclei (*Pinali and Kitmitto, 2014*; *Pinali et al., 2013*). While this organization may vary significantly from individual to individual, even in normal hearts, it ensures proper cardiac function. Multiscale studies of the heart, spanning whole organ to ultrastructural details, can reveal subtle deficiencies in CHD heart tissues, and relationships between abnormalities across spatial scales.

The correlative, multiscale imaging approach presented here was implemented and optimized in a chicken embryo model of heart development and CHD (*Midgett et al., 2017a*). Heart dimensions ranged from approximately 4–5 mm wide, 5-6 mm long, and 250–700 μm wall thickness. We acquired images of the whole embryonic heart using 3D micro-CT, and of cardiac ultrastructure using 3D SBF-SEM. Our multiscale imaging procedure achieved high-resolution images exhibiting both microstructural and ultrastructural preservation, while protocol completion was achieved in

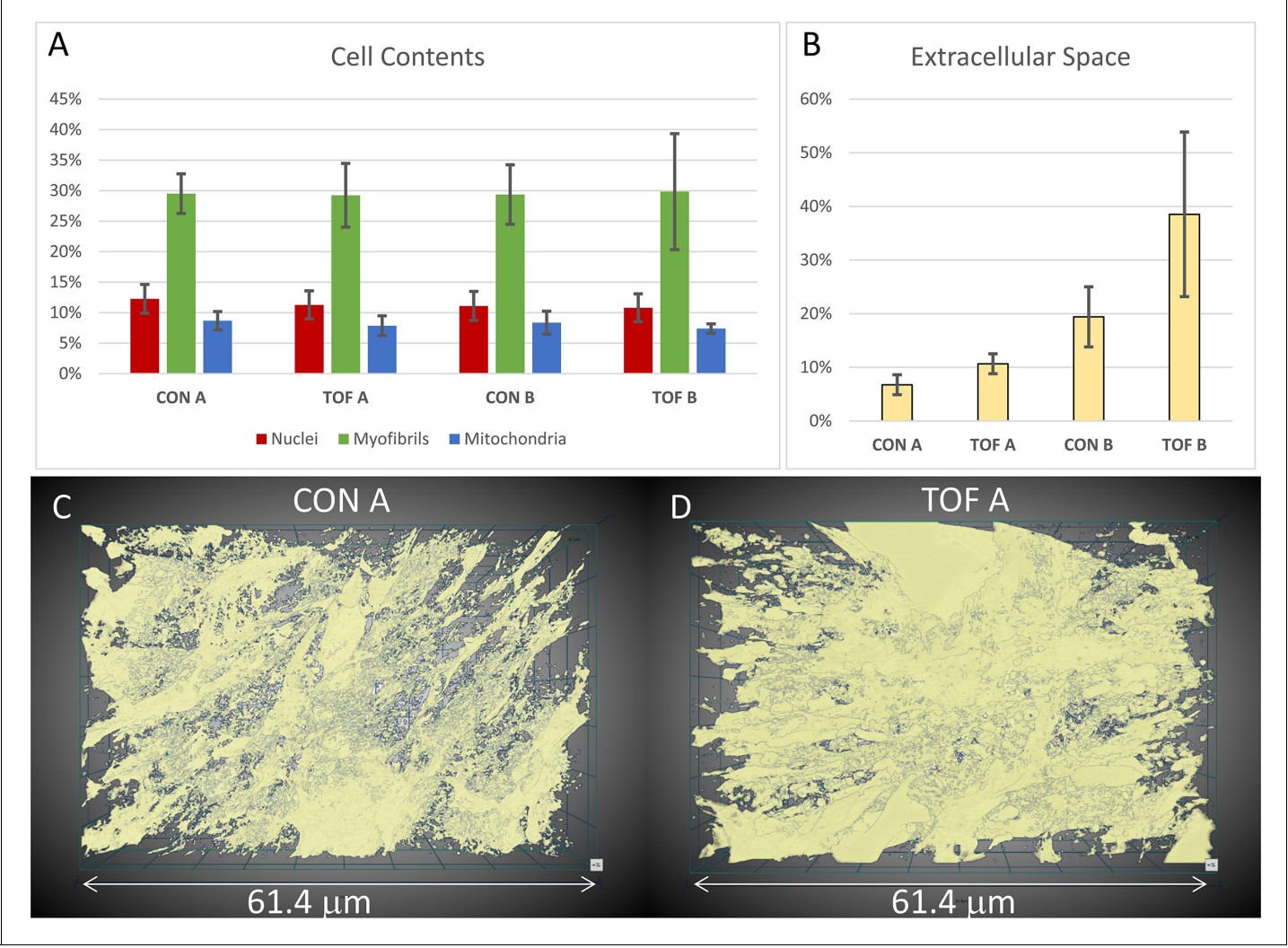

**Figure 8.** Segmentation and quantification of SBF-SEM images. Selected images (n ≥ 10) from the SBF-SEM image stacks acquired at regions A and B were segmented and quantified. (**A**) Percentage of myocardial cells occupied by nuclei, myofibrils and mitochondria in regions A and B of the control (CON) heart and TOF heart. (**B**) Percentage of the images occupied by extracellular space. (**C, D**) 3D views of the segmented extracellular space in (**C**) control (CON) heart, region A; and (**D**) TOF heart, region A. The semitransparent 3D views show increased extracellular space for the TOF A sample, as quantified in (**B**). Detailed of calculations presented in (**A**) and (**B**) are included as supplementary material (*Figure 8—source data 1*).

The online version of this article includes the following source data for figure 8:

**Source data 1.** Detailed calculations from segmented SBF-SEM images leading to *Figure 8A and B* plots.

about 4 days, which is comparable to completion timings for much smaller samples (*Genoud et al., 2018*). Our multiscale imaging methodology could enable studies of yet unknown tissue deficiencies in CHD.

The correlative multiscale imaging approach presented here could provide new insights into the underpinnings of heart development, and on cardiac mechanical changes in CHD. For example, analysis of multiscale cardiac images over developmental stages could be used to unravel how cardiac tissues progressively assemble and mature at diverse scales to form a functional heart. When cardiac development is perturbed (due to altered blood flow, hyperglycemia, hypoxia or any teratogen substance or changes in the environment) multiscale images can be used to reveal how the natural maturation and assembly of cardiac components (cells, extracellular matrix, organelles) are altered, sometimes in irreversible ways that lead to CHD. Multiscale images, furthermore, promise to reveal differences and similarities among CHD hearts with different phenotypes, as well as among hearts with similar phenotypes resulting from diverse insults. Moreover, the combination of morphological

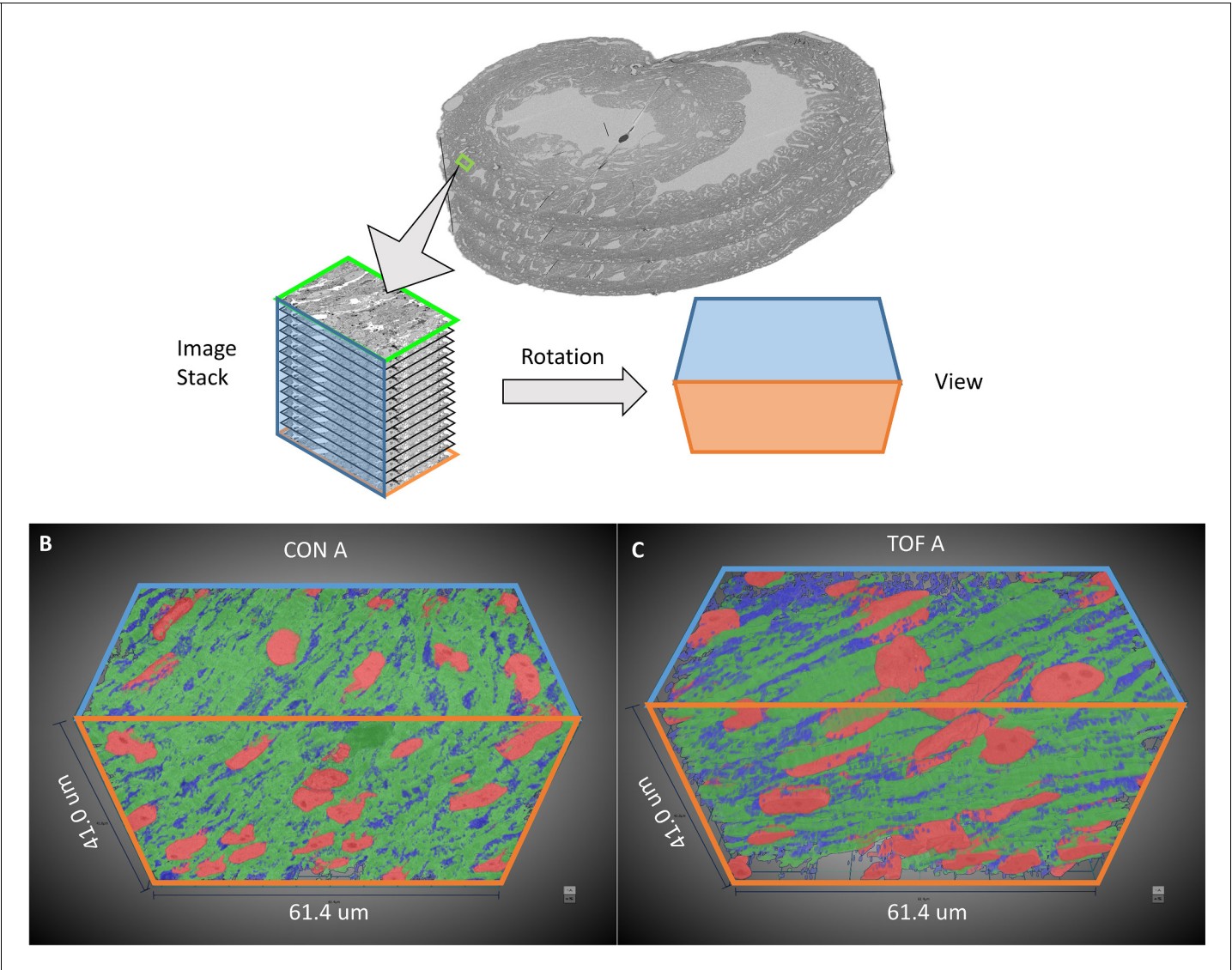

**Figure 9.** 3D visualization of SBF-SEM segmentations. (**A**) Sketch of image dataset acquisition, showing relative orientations. Green plane is the top image, orange plane is the bottom image (last image acquired), blue plane is approximately parallel to the heart wall. The sketch of the view shows the orientation of the planes as shown in (**B**) control (CON) heart and (**C**) TOF heart. Organelle segmentation is color-coded. Red: nuclei; Blue: mitochondria; Green: myofibrils.

and ultrastructural cardiac imaging data can predict the mechanical function of the heart. Indeed myocardial disarray or marked reductions in myofibrils, for example, will decrease the contraction efficiency (and force) of cardiac tissues. In turn, decrease contractility affects cardiac efficiency, including compromised ejection fraction and stroke volume.

## Our protocol in relation to previous works

Researchers have used EM techniques, including SEM, for decades to visualize the organization of organelles within cells (*Malick and Wilson, 1975*; *Karnovsky, 1965*). In the heart, studies using EM have revealed the ultrastructural architecture of mature myocardial cells (*Hussain et al., 2018*; *Pinali and Kitmitto, 2014*), and elucidated the maturation of myocardial ultrastructure during embryonic development (*Fischman, 1967*; *Wainrach and Sotelo, 1961*; *Manasek, 1970*). A few studies, moreover, have determined changes in ultrastructure due to pathophysiological conditions in the mature heart (*Sanchez-Quintana et al., 1999*; *Pinali et al., 2015*; *Holzem et al., 2016*).

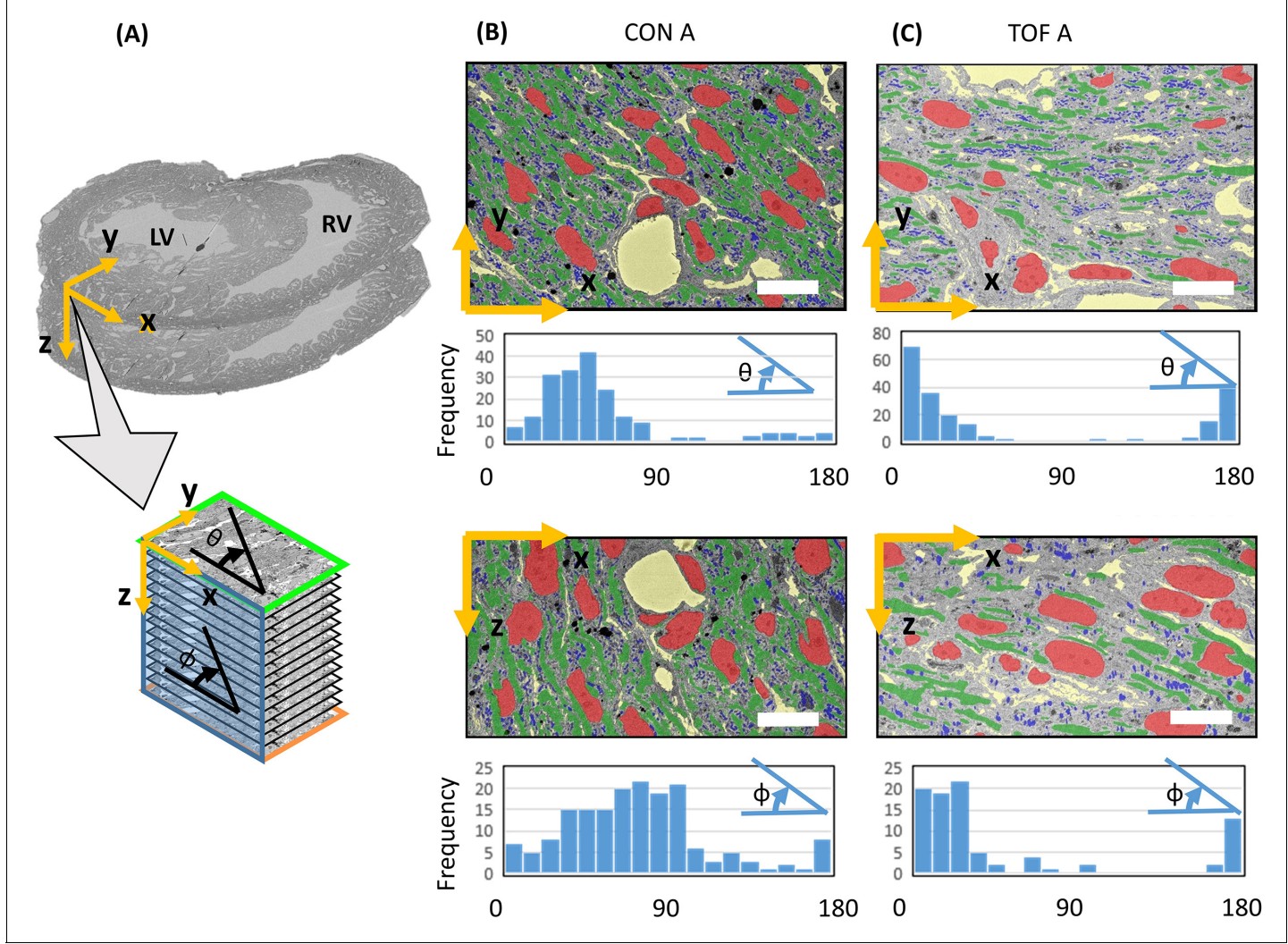

**Figure 10.** Myofibril transmural and elliptical angle quantification. (**A**) Sketch showing approximate position of the SBF-SEM imaged planes (x-y planes), and image stacks (spanning the z-direction depth) with respect to the heart morphology. For the imaged transmural planes, the y-direction approximately corresponds to the radial direction in the left ventricle (LV), and the x-direction is approximately parallel to the LV wall boundary (circumferential direction). The stack depth direction (z-direction) corresponds to the heart longitudinal direction. Myofibril orientation angles were quantified for the x-y and x-z planes: the transmural angle θ is the myofibril angle in the x-y plane, with respect to the circumferential (x) direction (wall direction); the elliptical angle Φ is the orientation angle in the x-z plane with respect to the circumferential (x) direction. Segmentations of organelles on the x-y and x-z planes, together with frequency versus angle histograms, are shown in (**B**) for the control heart, ROI A (CON A); and (**C**) for the TOF heart, ROI A (TOF A). Each bar in the histogram represents a 10° range, hence there are 18 bars to represent 0° to 180° angles. Organelle segmentation is color-coded. Red: nuclei; Blue: mitochondria; Green: myofibrils. The extracellular matrix is shown in yellow. Scale bars: 10 μm. The details of the angle calculations are included as supplementary files (*Figure 10—source datas 1–4*).

The online version of this article includes the following source data for figure 10:

**Source data 1.** Details of transmural angle θ calculation from SBF-SEM images of the control heart CON A sample.
**Source data 2.** Details of elliptical angle Φ calculation from SBF-SEM images of the control heart CON A sample.
**Source data 3.** Details of transmural angle θ calculation from SBF-SEM images of the TOF A heart sample.
**Source data 4.** Details of elliptical angle Φ calculation from SBF-SEM images of the TOF A heart sample.

Properly preparing samples for EM requires meticulous protocols that aim at preserving the ultra-structure of the tissue under study (e.g. intact cell and nucleus membranes, mitochondria and their crystae, myofibrils and z-disks). Because the ultrastructural features analyzed are at the nanometer scale, samples used for EM are typically very small (<1 mm³), which facilitates proper sample prepa-ration. In preparing samples, portions of the heart are typically excised, carefully prepared for

imaging (fixed and stained) (*Rennie et al., 2014*; *Mukherjee, 2016*), and then imaged with an EM modality (*Rennie et al., 2014*; *Pinali and Kitmitto, 2014*). Using this procedure, however, finding the ultrastructure associated with a specific microscopic feature or malformation site can be daunting. Moreover, in micro-dissecting tissue samples, the myocardial organization within the heart may be lost (*Gilbert et al., 2007*). Our methodology enables correlative microscopy in a way that allows precise identification and mapping of portions of the heart to their ultrastructure.

Whole animals and organs have been scanned with computed tomography (CT), a 3D X-ray imaging modality, to determine the internal and external structure of organs, including the heart. For small animal models, micro-CT, a high-resolution CT, is typically employed (*Midgett et al., 2017a*; *Butcher et al., 2007*). To enhance contrast and thus resolution, prior to micro-CT imaging excised tissue samples are stained (*Midgett et al., 2017a*). Preparing samples for micro-CT imaging requires good and uniform penetration of the stain, intended to preserve and contrast the tissue microstructure (e.g. heart morphology, heart chambers and valves). Micro-CT can then reveal subtle and overt malformations in the heart and its microstructure (*Midgett et al., 2017a*). For example, using micro-CT cardiac images it is possible to visualize ventricular septal defects or translocations of the great arteries, but also wall and septum thickness, and differentiate trabecular from compact myocardium (*Midgett et al., 2017a*).

The discrepancy in the scale at which we acquire micro-CT and SBF-SEM images introduces fundamental differences in the requirements for sample preparation. Due to diverse fixation and staining protocols (*Midgett et al., 2017a*; *Hopkins et al., 2015*), tissues prepared for micro-CT or other microstructural imaging (e.g. histology) cannot typically be simultaneously processed for SBF-SEM (and other EM modalities), thereby restricting the ability to analyze both the microstructural and ultrastructural characteristics within the same tissue sample. Our correlative micro-CT/SBF-SEM procedure can reveal the sub-cellular architecture associated with specific pathological or malformed regions of the heart found from micro-CT images.

Applications combining micro-CT and EM technologies have recently begun to emerge, for example (*Karreman, 2017*; *Morales et al., 2016*; *Sengle et al., 2013*). However, several challenges remain in applying these methods to correlative, multiscale imaging of a relatively large organ like the heart (even the heart of a small animal). To achieve both EM and micro-CT high-quality imaging of the same sample, existing protocols have capitalized on heavy metal contrast in small tissue samples, which are fully processed prior to EM and micro-CT imaging, for example (*Genoud et al., 2018*; *Karreman, 2017*). However, achieving the uniform staining necessary for optimal imaging with both micro-CT and SEM becomes progressively challenging with increasing tissue sample size. This is mainly due to difficulties in achieving uniform and fast fixation (that preserves the ultrastructure), and uniform stain penetration (both for post-fixation purposes and to enhance contrast for SEM and micro-CT imaging). Modifications to the classic ROTO protocols (*Hua et al., 2015*; *Malick and Wilson, 1975*; *Willingham and Rutherford, 1984*) for EM tissue preparation have been quite successful in achieving strong and uniform staining of relatively large samples (*Deerinck et al., 2010*; *Tapia et al., 2012*). However, acceptable staining was typically only up to a depth of 200 µm, and more recently 500 µm (*Hua et al., 2015*) in dense brain tissues. In an attempt to stain whole brains for EM reconstruction of synapses, Mikula et al. developed the brain-wide reduced-osmium staining with pyrogallol-mediated amplification (BROPA) protocol (*Mikula and Denk, 2015*) for 3D SBF-SEM (no other heavy metals were used). While the protocol is compatible with both micro-CT and 3D SBF-SEM imaging, preparing a whole mouse brain (about 8–10 mm in diameter) using BROPA required 2–3 months. A fast BROPA protocol (fBROPA) was later developed and used to prepare whole brains from zebrafish in about 4 days (*Genoud et al., 2018*). Zebrafish brains, however, are significantly smaller than mouse brains (diameter of 1.1 mm vs 8–10 mm, respectively). For our hearts, we needed to achieve stain penetration of a relatively large sample (4–5 mm wide) and a fast preparation protocol was also desired. We found that preparing the heart for SBF-SEM and stopping the protocol after initial ROTO staining (1 day processing), was compatible with micro-CT and later SBF-SEM full sample processing and imaging. Further, full sample preparation compatible with SBF-SEM required and additional 3 days, making the entire heart processing around 4 days. To our knowledge, this is the first time that correlative micro-CT/SBF-SEM imaging is applied to the heart.

## Protocol implementation

Sample preparation for SBF-SEM required strong, immediate fixation to preserve the ultrastructure of the heart tissue. We used a modified Karnovsky's fixative with equal parts glutaraldehyde and paraformaldehyde. The paraformaldehyde rapidly penetrated and temporarily stabilized the tissue, and the slower-penetrating glutaraldehyde, a superior cross-linker, more permanently preserved the tissue sample (*Karnovsky, 1965*). An obvious difficulty was to achieve uniform fixation of the whole heart sample. Homogenous fixation was achieved by perfusing fixative into the heart prior to excision and then promptly immersing the heart in fixative after excision, allowing the fixative to simultaneously penetrate the heart through the tissue's internal and external surfaces. The hearts were then post-fixed with osmium tetroxide, a lipid cross-linker, which fully stabilized membrane structures while enhancing contrast for micro-CT and SBF-SEM imaging. For our heart samples, we could achieve uniform stain penetration using a variation of the ROTO protocol with extended staining timing (about 30% increase; see Materials and methods). The extended timing was sufficient for our hearts, even considering small size variations. We expect, however, that further time increases would apply to larger heart samples (for instance if we image embryos at a more advanced developmental stage, or other species are considered). For the chick embryos studied here, sample preparation after ROTO was adequate for micro-CT imaging of the whole heart (see *Figure 2*).

For 3D SBF-SEM images, heart samples were further stained with a combination of heavy metals. This is because SEM images are acquired via the detection of secondary and backscattered electrons that are emitted as the tissue is scanned with a high energy beam of primary electrons (*Cazaux, 2012*). Soft biological tissues, like the heart muscle, yield few backscattered electrons and need to be stained with heavy metals, which readily produce secondary and backscattered electrons. Heavy metal stains interact with specific ultrastructural components, and therefore combinations of stains are frequently used in a single sample. The application of osmium tetroxide, used in ROTO protocols, served as the first application of a heavy metal stain in the SBF-SEM preparation. The osmium tetroxide, which interacts with lipids in membranes and vesicles, both post-fixed and stained tissues. Further staining with uranyl acetate stained lipids and proteins, and lead aspartate stained proteins and glycogens. Together, these heavy metal stains fully preserved and contrasted ultrastructural details, as evidenced by high-resolution SBF-SEM images (see *Figure 5*).

For whole-heart samples, we found that staining with uranyl acetate and lead aspartate rendered resin-embedded hearts opaque to micro-CT imaging (data not shown). Our multiscale multimodality imaging procedure overcame these difficulties by performing micro-CT imaging after ROTO (see above) but prior to the lead and uranium staining steps, such that correlative 3D micro-CT and 3D SBF-SEM could be implemented. This initial modified ROTO post-fixing and staining provided tissue contrast for micro-CT while also ensuring that the sample was preserved and stabilized before final SBF-SEM sample preparation and imaging (see *Figure 1*). In addition, ROTO post-fixing enabled screening, storing and selection of hearts before final SBF-SEM processing. This feature of our multiscale approach is advantageous for several reasons. At this early processing point, hearts could be stored for relatively long periods (at least 2 weeks, but potentially months/years) before further processing. This allows researchers to prepare and screen by micro-CT a large number of hearts, and then select only those hearts of interest (e.g. with a specific malformation) for further analysis with SBF-SEM. Not only does this approach permit banking of samples, but it also saves considerable time and resources by avoiding full sample preparation of hearts that are not useful. This is important for our application, in which at most 60% of treated hearts develop structural malformations, and the nature and severity of defects vary among individual hearts. Since we cannot accurately classify malformations until they are scanned with micro-CT, being able to use the micro-CT as both a screening tool and as a navigational tool for later correlative microscopy is invaluable to CHD studies.

Due to the size of the hearts and slow diffusion (penetration) of heavy metals into the tissue, achieving uniform staining during further SBF-SEM tissue processing was not straightforward. Initial iterations of the procedure (data not shown) resulted in a strong gradient of staining into the heart tissue, a manifestation of poor stain penetration (*Genoud et al., 2018*; *Mikula and Denk, 2015*). Application of microwave steps to enhance diffusion was not helpful, although perhaps optimizations of those steps could achieve improved results. We found, however, that

following a Renovo Neural, Inc protocol (see Materials and methods) after ROTO and before embedding the sample in a resin capsule allowed the samples to achieve homogeneous staining. Uniform stain penetration throughout the heart, was apparent from semithin transverse sectional images of the heart and low-resolution images of ROIs (see *Figure 4*). The uniform contrast and resolution of ultrastructural details provided further evidence of uniform staining and proper tissue fixation (see *Figure 5*).

While the focus of this study was to demonstrate homogeneous staining and fixation, our procedure enables correlative microscopy. One way to achieve accurate localization of ultrastructures within the heart structure, is to register semithin transverse sections to micro-CT images, and then SBF-SEM images to the semithin images (as done in *Figure 4*). Because sectioning of samples for SBF-SEM imaging is done after micro-CT images are acquired, sectioning is guided by the images of the whole heart, facilitating the selection of regions of interest. Registration can then be performed among the images themselves. This could be done directly, or by adding fiduciary markers in the resin/heart to facilitate image alignment. Our procedure allows imaging of ultrastructure at several regions of interest within the heart, enabling extensive ultrastructural mapping.

## Comparison of control and TOF hearts and limitations of this study

We explored some possible analysis and quantification strategies enabled by our multiscale imaging procedure. We acknowledge that results from this study are very preliminary: Analysis of more heart samples is needed to reach conclusions applicable to CHD. In the future, combining echocardiography, which can acquire in vivo images of the heart for functional analysis (including blood flow) (*Midgett et al., 2017a*), together with 3D micro-CT and SEM, can reveal functional as well as detailed microstructural and ultrastructural characteristics of normal versus CHD hearts. Further, a combination of segmentation, quantification and other refined methods to interrogate images (at the functional, microstructural and ultrastructural levels) will elucidate similarities and differences between normal and malformed hearts, possibly informing therapeutic treatment strategies. To be biologically meaningful, however, these studies need to include more animals. The quantifications and comparisons presented here for one control and one TOF heart (thus n = 1) pertain only to these two hearts, and are presented as an illustration of possible ways of extracting information from the proposed multiscale imaging method.

Micro-CT images reveal the microstructural characteristics of hearts. Not only could we classify the hearts based on phenotype (normal vs TOF), but cardiac characteristics, such as heart size and wall thickness could be visualized and quantified. For example, it was noted from our analysis that the RV of the TOF heart exhibited a larger volume than that of the CON heart examined here (see *Figure 3*). During fetal life, the lungs are not functional, and blood to the lungs is shunted to the systemic circulation through the ductus arteriosus (*Kiserud, 2005*) (a pair of ducti in chick [*Dzialowski, 2018*]). The RV hypertrophy characteristic of TOF, develops over time after the baby is born (*Iacobazzi et al., 2016*). RV hypertrophy, therefore, may not be present at the fetal stages of heart development examined in this study and was not observed in our TOF heart. In the future, it would be interesting to determine whether the trait observed in this study is preserved among TOF fetal hearts, and if so under which conditions and how it affects the RV wall ultrastructure (not examined here). To be meaningful, however, such study requires a larger number of heart samples, and is outside of the scope of this paper.

Another difference between the two hearts was that the ventricles of the TOF heart exhibited less dense tissue and a more extended trabecular architecture compared with the control heart. This is consistent with reduced myocardial compaction in TOF (*Jenni et al., 1999*; *Weiford et al., 2004*; *Finsterer et al., 2017*). The heart trabeculae is characterized as a 'spongy' or porous tissue that develops inside the heart ventricles, and in our samples was evident from semithin transverse heart sections (*Figure 4*), but could also be approximately quantified as the extracellular portion of the images (*Figure 8*). It has been shown that the heart trabecular architecture is sensitive to blood flow conditions during development (*Sedmera et al., 1999*), and thus a disrupted trabecular architecture may be a characteristic of many CHD hearts due to their anomalous flow characteristics during fetal stages (*Wiputra et al., 2018*). However, the trabecular and myocardial architecture can also exhibit variations from heart to heart (*Gilbert et al., 2007*), therefore further analysis with a larger sample size is required before we can make conclusions related to TOF.

We noticed differences in SBF-SEM image sharpness, which are attributable to excessive 'charging' (accumulation of static charge on a sample's surface) when scanning the TOF heart (*Figure 5*). The increased charging in our TOF heart is linked to its trabeculation, which features larger and more numerous void regions filled with free resin. To address this problem during imaging, we slightly shortened the dwell time when acquiring SBF-SEM images from the TOF sample. In the future, we could embed silver particles in the resin to increase sample conductivity and enhance image quality (*Genoud et al., 2018*). Nevertheless, the image quality of both the TOF and CON hearts was sufficient to appreciate ultrastructural details (*Figure 5*).

While 2D EM images have been invaluable in deciphering ultrastructural features of myocardial cells and tissues, 3D images can unravel more details in the spatial organization of the ultrastructural architecture (*Hussain et al., 2018*; *Pinali et al., 2015*). As an example, segmentation and quantification of the 3D data revealed that myofibril alignment was slightly different between our TOF and control hearts (*Figure 10*). This is perhaps because the ROIs from the two hearts are not exactly corresponding, or due to the more extended trabecular architecture of the TOF heart, and warrants further investigation. For myocardial alignment quantification, it is also important to arrest the heart consistently (in diastole as done here, or systole) as myocardial cell orientation changes over the cardiac cycle (*Omann et al., 2019*; *Sonnenblick et al., 1967*). 3D SBF-SEM images also revealed a greater proportion of endocardial cells in TOF heart tissues than in control tissues, such that volumetric studies not focusing on myocardial cells show reductions in the myofibril density of the TOF heart (data not shown). When the analysis was focused exclusively on myocardial cells, however, we could not find any differences in the density of myofibrils or mitochondria (*Figure 8*). Outside the scope of this paper, but relevant to the comparison of normal versus CHD hearts, would be an extensive analysis of left and right ventricular wall microstructure and ultrastructure, including the distribution of lipid droplets, glycogen, and mitochondria with respect to the myofibrils. In addition, studies of the ultrastructural organization within myocardial, endocardial, fibroblast and conduction cells, in normal and CHD hearts, would be relevant to decipher the impact of CHD on cell and cardiac function. While outside the scope of this paper, future studies that include more animals should focus on elucidating ultrastructural cardiac differences in animal models of CHD as such differences can inform clinical studies and impact the lives of children and adults with congenital heart defects. Our proposed multiscale imaging methodology could certainly enable such studies.

## Studies enabled by multiscale imaging

Previous studies have detailed the congenital heart anomalies associated with outflow tract banding (OTB) *Gessner, 1966*; *Clark et al., 1984*; *Midgett et al., 2014* used in this study to induce TOF. Those studies found a spectrum of congenital heart defects after OTB, which originated from abnormalities in the outflow tract (conotruncal defects). These abnormalities included increased separation between the aortic and mitral valve annuli (altered aortic-mitral valve continuity), ventricular septal defects, abnormal position of the aorta, including TOF and double outlet right ventricle (DORV), in which both the aorta and pulmonary trunks emerge from the right ventricle. Meanwhile, neural crest cell ablation also leads to conotruncal anomalies in the chick embryo (*Hutson and Kirby, 2003*; *Kirby et al., 1983*). Cardiac neural crest cells are required for normal heart development (in the chick and mouse), and ablation of these cells leads to persistent truncus arteriosus (PTA), characterized by lack of separation of the aorta and pulmonary trunk, but also to TOF and DORV. Likewise, diverse genetic anomalies are also associated with conotruncal heart defects (*Srivastava and Olson, 2000*; *Rugonyi, 2016*). However, the mechanisms by which anomalous genes, neural crest cell ablation, and altered hemodynamics lead to conotruncal defects may differ, and these differences may impact myocardial and myofiber orientation and maturation. Multiscale imaging studies could reveal the impact of diverse interventions on cardiac microstructure and ultrastructure, both when the same or different phenotypes are obtained. This in turn could contribute to our understanding of the underpinnings of CHD and their functional and structural consequences.

Direct application of the proposed multiscale imaging method to human hearts is limited. The method presented here is destructive, and thus can only be applied to human samples of deceased individuals. Moreover, the larger size of the human heart will lead to difficulties in attaining homogeneous fixation and staining. To circumvent fixation and stain homogeneity issues, increased timing for diffusion of fixative and heavy metal stains is certainly a possibility as is microwave steps (to

accelerate diffusion). In addition, changes in processing, such as sectioning of hearts after micro-CT to facilitate diffusion of heavy metal stains and control imaging are also possible. As presented, using our methods, multiscale imaging of human hearts is limited.

## Conclusions

Our correlative, multiscale imaging procedure allowed us to acquire detailed micro-CT images of an entire embryonic chicken heart (see *Figure 2*), followed by ultrastructural 3D SBF-SEM images from the same heart (see *Figures 5* and *6*). The described approach allows the correlation of microstructural and ultrastructural architecture in selected regions of the heart. This is important when studying CHDs, as each malformation phenotype may be different and therefore may need to be analyzed separately to fully appreciate multiscale effects and to understand how phenotypes affect cardiac architecture at disparate levels. Furthermore, similar phenotypes that result from diverse insults (e.g. hemodynamics, neural crest cell ablation) could also lead to dissimilar cardiac ultrastructure and function. Importantly, multiscale studies can be used to decipher the imprints that early alterations in the environment in which the heart is growing have on cardiac formation and function. Other potential applications to CHD (and beyond) are determinations of extracellular matrix organization/disorganization, cardiac fibrosis, glycogen distribution and myxomatous degeneration of valve tissues in response to different insults and aging. The multiscale imaging approach presented here therefore could enable animal studies to inform how human cardiac anomalies, even when repaired, could subsequently lead to increased cardiac dysfunction and heart failure. For patients with CHD, such studies may further reveal associated pathologies in cardiac tissues that, if not properly treated, may have devastating implications for survival and long term cardiac health.

While our multiscale imaging approach was implemented and optimized using embryonic chicken hearts, we expect it will be straightforward to adapt it for use in mouse and other small animal models of cardiac malformations. It will be advantageous to use complementary models, as typically genetic insults are studied using mouse models, while environmental perturbations are studied using avian models. Heart dimensions in those species (mouse and chicken) are very similar, and we anticipate that tissue processing will not differ significantly. Slight increases in heart size may just require an increase in protocol staining times. Extending the approach to different species and models of congenital heart disease will likely enable us to understand in detail the similarities and differences between cardiac defects, and the underpinnings of malformations that result from genetic and environmental insults.

## Materials and methods

**Key resources table**

| Reagent type (species) or resource | Designation | Source or reference | Identifiers | Additional information |
|---|---|---|---|---|
| Chemical compound, drug | Sodium Cacodylate | EMS | Cat#RT 12300 | RT 12300 0.1M; pH 7.4 |
| Chemical compound, drug | Tannic acid | EMS | Cat#21700 | 0.1% (w/v) |
| Chemical compound, drug | Osmium Tetroxide, $OsO_4$ | Ted Pella | Cat#18463 | 2% (v/v) final concentration |
| Chemical compound, drug | Potassium Ferricyanide, $K_3[Fe(CN)_6]$ | EMS | Cat#20150 | 1.5% (w/v) |
| Chemical compound, drug | Thiocarbohydrazide, TCH | Aldrich | Cat#223220 | 0.1% (w/v) |
| Chemical compound, drug | Uranyl acetate | Ted Pella | Cat# 19481 | 1% (w/v) working solution |
| Chemical compound, drug | Lead nitrate | EMS | Cat#17900 | Used to prepare Lead Apartate |

*Continued on next page*

*Continued*

| Reagent type (species) or resource | Designation | Source or reference | Identifiers | Additional information |
|---|---|---|---|---|
| Chemical compound, drug | Aspartic acid | Aldrich | Cat#11195 | Used to prepare Lead Apartate |
| Chemical compound, drug | Acetone | EMS | Cat#10015 | Glass Distilled |
| Chemical compound, drug | Epoxy resin | EMS | Cat# RT 14900 Epon 812 | |
| Software, algorithm | Amira 6.0 | FEI, now ThermoFisher | | |
| Software, algorithm | Dragonfly 4.1 | Object Research Systems | | |

## Ethical considerations

Our research used chicken embryos. According to the US National Institutes of Health (NIH) Office of Laboratory Animal Welfare (*ILAR News* 1991; 33(4):68–70), the NIH's 'Office for Protection from Research Risks has interpreted 'live vertebrate animal' to apply to avians only after hatching.' Our Institutional animal care and use committee (IACUC) follows NIH interpretation. Therefore, chicken embryos are not considered animals and our research did not require approval. Incubator logs in the lab were monitored daily to ensure there were no eggs near the hatching time of 21 incubation days. Nevertheless, we used the minimum possible number of embryos to achieve our goals.

## Generation of cardiac defects

Our multiscale approach was implemented and optimized using fully formed embryonic chicken hearts (heart length ~ 5–6 mm), and applied to a chick animal model of congenital heart disease. Chicken embryos were prepared as described previously (*Midgett et al., 2017a*). Briefly, fertilized white Leghorn chicken eggs were incubated blunt end up at 38°C and 80% humidity for approximately 3 days (to Hamburger and Hamilton (HH) stage HH18 [*Hamburger and Hamilton, 1992*]). Control and treatment interventions were then performed as described below and the embryos were re-incubated for an additional 9 days (to HH38, when the heart has four chambers and valves). Two embryonic hearts were included in this study: (1) a control, normal heart; and (2) a malformed heart with tetralogy of Fallot (TOF). TOF was achieved by performing outflow tract banding (OTB) at HH18, wherein a 10–0 nylon suture was passed under the mid-section of the heart outflow tract and tied in a knot (band tightness 38%). The band was removed from the outflow tract ~ 24 hr after placement (HH24), and then the embryo was allowed to develop to HH38. The control heart was obtained by passing a 10–0 nylon suture under the heart outflow tract without knotting it, and subsequently allowing the embryo to develop to HH38. Embryo hearts were collected at HH38 for multiscale imaging.

## Homogenous fixation of whole hearts

At HH38, embryonic whole hearts were excised and fixed as follows. The chest cavity was opened and the pericardial sac around the heart gently removed with forceps. Each heart was arrested by injecting 500 μL of chick ringer solution containing 60 mM KCl, 0.5 mM verapamil, and 0.5 mM EGTA (*Tobita et al., 2005*) into the left ventricle through the heart's apex. Hearts were then immediately perfused with ~2 mL of ice-cold (0°C) modified Karnovsky's fixative (2.5% Glutaraldehyde and 2.5% PFA in PBS (pH 7.4)) through the same injection site. All perfusions were performed with a 21 gauge needle. A transfer pipette was used to quickly apply ~ 1 mL of fixative to the heart's exterior to ensure uniform fixation of the heart tissue. Next, the heart great vessels were cut with small spring scissors and hearts were placed in 1.5 mL fixative and stored at 4°C until further processing.

## Cardiac processing enabling micro-CT imaging

In order to enable both whole-heart micro-CT imaging and subsequent SBF-SEM imaging of regions of interest (ROIs), we processed fixed hearts for micro-CT using the initial portion of a Renovo Neural, Inc (Cleveland, USA) protocol (*Mukherjee, 2016*) designed for SBF-SEM imaging (see *Figure 1*,

*Step 1*). Each heart was placed in a 5 mL glass scintillation vial and we used 3 mL of solution per vial for each incubation/wash. First, the fixed hearts were washed in 0.1M Sodium Cacodylate (pH 7.4) for 20 min with 4 exchanges of fresh buffer. Next, the hearts were incubated in 0.1% (w/v) of tannic acid in 0.1M Sodium Cacodylate (pH 7.4) for 15 min at room temperature. Samples were then washed in 0.1M Sodium Cacodylate (pH 7.4) for 20 min with 4 exchanges of fresh buffer. Since the reducing agents used in subsequent steps (modified ROTO protocol) were light-sensitive, the sample vials were covered in aluminum foil from this point on. The whole hearts were post-fixed in 2% (v/v) Osmium Tetroxide ($OsO_4$) and 1.5% (w/v) Potassium Ferricyanide ($K_3[Fe(CN)_6]$) in distilled water ($dH_2O$) for 2 hr at room temperature on a rotating platform. The samples were then extensively washed in $dH_2O$ for 20 min with four exchanges of fresh $dH_2O$. Next, the samples were immersed in 0.1% (w/v) Thiocarbohydrazide (TCH) solution in $dH_2O$, placed in an oven, and incubated for 40 min at 60˚C. This step was followed by another four exchanges of fresh $dH_2O$ over 20 min. Samples were then immersed in a 2% (v/v) $OsO_4$ solution in $dH_2O$ for 2 hr at room temperature on a rotating platform. Finally, the hearts were washed extensively in $dH_2O$ over 20 min with four exchanges of fresh water. Each heart was stored in $dH_2O$ at 4˚C until imaged by micro-CT. This preparation provided excellent contrast for micro-CT scans (see Results).

Micro-CT images of whole hearts were acquired to assess the cardiac structure. We acquired high-resolution (~10 µm) 3D scans of each heart using a Caliper Quantum FX Micro-CT system (Perkin-Elmer, CLS140083) with 10 mm field of view, 140 µA current, 90 kV voltage, and a scan time of 3 min. We used the Amira 6.0 software platform (FEI Company) or Dragonfly 4.1 software (Object Research Systems, Quebec, Canada) to visualize these scans and identify cardiac defects. Hearts were then stored in double distilled water at 4˚C until further processing. Please note that at this step in the processing (cardiac tissues fixed and post-fixed with $OsO_4$) water does not damage the tissues.

## Subsequent cardiac tissue processing enabling 3D SBF-SEM imaging

After whole hearts were imaged with micro-CT, sample preparation of selected hearts for 3D SBF-SEM imaging was finished (see *Figure 1*; *Step 2*), following the Renovo Neural, Inc protocol. In large samples, like the whole hearts described in this manuscript, it is necessary to extend most of the staining steps. Failure to extend the timing of staining resulted in a heterogenous stain distribution throughout the tissue (in our early iterations of the procedure). In our final, optimized procedure, we incubated the samples in 1% (w/v) aqueous uranyl acetate for 24 hr at 4˚C, after which they were washed in $dH_2O$ for 30 min with 6 exchanges of fresh $dH_2O$. We then incubated the samples in lead aspartate for 30 min at 60˚C. The samples were then extensively washed in $dH_2O$ for 20 min with 4 exchanges of $dH_2O$. Dehydration steps were done in a series of acetone-$dH_2O$ mixtures (50, 75, 85, 95, and 100%); each step was repeated twice for 5 min at room temperature. The whole heart sample was then embedded in an epoxy (Epon 812) resin for further manipulation and SBF-SEM sample preparation. The first infiltration step was done for 1 hr at room temperature in a mixture of 1:1 (v/v) acetone:epon followed by a 1:3 (v/v) acetone:epon incubation for 1 hr at room temperature. The hearts were subsequently incubated overnight in pure (100%) epon on a rotating platform. The following day the epoxy solution was exchanged four times, each time with 30 min incubation steps at room temperature. Samples were polymerized at 60˚C for 48 hr in a conventional oven, leading to a whole heart sample embedded in an Epon block.

## Selection of regions of interest (ROIs) and SBF-SEM image acquisition

Using the micro-CT images as reference, the Epon-embedded heart blocks were sectioned to reach a selected short axis (transverse) section using a diamond-wire jewelry saw. For this study we selected the mid cardiac transverse section, at a plane where the heart is wider (the equatorial plane). After this step, a semithin section (250 nm) was obtained using an ultramicrotome and mounted on a silicon chip previously glow discharged for 1 min at 15 mA (PELCO easyGlow, Ted Pella). Semithin section images were used to confirm the area of interest as well as to check for both the ultrastructural quality of the sample and the success of the staining procedure (see *Figure 1*; *Step 2*). This step is crucial since the SBF-SEM imaging requires samples with extremely good contrast. Semithin sections were imaged on a Teneo Volume Scope in low vacuum mode using a VS-DBS backscattered electron detector and the MAPS software (FEI Company). Imaging conditions

used were 2.5 kV and 0.2 mA, dwell 3–5 µs. In some cases, the samples imaged using this method needed to be coated with a thin (5–8 nm) layer of carbon to minimize charging artifacts induced by the electron beam.

The same diamond-wire jewelry saw was then utilized to generate a slab (~1.5 mm) from the sample (see *Figure 1*; *Step 3*). The slabs were sectioned into smaller ROIs, which were subsequently mounted on Microtome stub SEM pins (Agar Scientific 61092450) using H20E Epo-Tek silver epoxy (Ted Pella 16014) and cured overnight at 60˚C in a conventional oven. The resulting small blocks were then trimmed using a Trim90 diamond knife (Diatome) to generate a pillar of $500 \times 500$ µm$^2$. The block was then coated with 20 nm of gold using a Leica ACE 600 unit.

In the last step of our multiscale imaging procedure, 3D SBF-SEM images of sub-ROIs selected from the mounted sample were acquired (see *Figure 1*; *Step 4*). 3D image acquisition was done on a Teneo Volume Scope SBF-SEM in low vacuum mode (50 Pa) using a VS-DBS backscattered detector. Images were acquired at a lateral resolution of 10 nm/pixel and image sets included 800–1000 serial sections (with each section thickness measuring 40 nm in the z axis). SBF-SEM data sets were approximately 40 µm $\times$ 60 µm$\times$32–40 µm.

## Image analysis and segmentation

All registration, of SBF-SEM data was performed with Amira 6.0 (FEI Company). First, complete image stacks (800–1000 slices) from each ROI and sub-ROIs were automatically aligned to generate a continuous 3D volume. Next, a non-local means filter was applied to every 2D slice in order to improve the signal-to-noise ratio. Due to slight differences in the intrinsic properties of the tissue, sections from the TOF heart appeared slightly lighter compared to the control heart. We adjusted the intensity of the TOF sections during post-processing to match that of the control heart.

To better appreciate ultrastructural differences between the two hearts, we used Dragonfly 4.1 software to segment and quantify SBF-SEM images. We segmented: cell nuclei, mitochondria, myofibrils and the extracellular space (this later one to allow quantification of relative organelle volume within cells). The segmentation used a combination of tools in Dragonfly, including deep learning algorithms. Briefly, we employed a six-level U-Net deep learning model (*Ronneberger et al., 2015*) implemented in Dragonfly to perform an initial segmentation of nuclei, mitochondria, and myofibrils within imaged cells. Training sets required for the deep learning model were obtained initially through manual segmentations of a few selected images from the image stacks. The training set was later augmented by applying the segmentation deep learning algorithms to other (selected) images from the set, follow by thorough manual cleaning. For each training session, the model was run for 50 epochs with a patch size of 128 pixels. Dragonfly automatically divides the training sets into training and validation regions, so that training (and further inclusion of training images) continued until the reported accuracy (from validation regions) was > 98% with a loss < 0.06. Images were then segmented with the trained model, and segmentations further refined both using Dragonfly automated tools, such as morphological operations, and manual clean up using painter tools available in Dragonfly. Unlike organelles, the extracellular space was easily recognized by intensity levels, and thus simply segmented based on its intensity, followed by clean up using both automatic and manual tools in Dragonfly.

For visualization purposes, we segmented a whole dataset from the control heart and an approximately corresponding dataset from the TOF heart (region A, *Figure 9*). The total volume of the dataset was $40 \times 60 \times 32$ µm$^3$. For quantification purposes, we segmented and further curated smaller portions of the data sets from two corresponding regions of the control and the TOF hearts (regions A and B). Quantifications of extracellular space and nuclei were done from 17 evenly spaced images from the 800 image datasets of region A; and 21 evenly spaced images from the 1000 image datasets of region B (thus every 50th image was used for these segmentations). For quantification of mitochondria and myofibrils, we cropped images (n $\geq$ 10) so that we could focus on smaller regions, allowing us to manually improve the accuracy of segmentations in a more tractable manner and focusing on myocardial regions.

Quantifications were performed based on segmented images. We quantified, from each image or image portion, the total surface area ($S_T$), and the surface area occupied by extracellular space ($S_E$), nuclei ($S_N$), mitochondria ($S_{Mit}$) and myofibrils ($S_{Myo}$). We then computed the fraction of the total surface area occupied by the extracellular space ($S_E/S_T$); and the fraction of the cell occupied by organelles (nuclei, mitochondria and myofibrils), computed as the ratio of organelle surface ($S_i$, with i = N,

Mit, Myo) to the cell surface ($S_i/(S_T - S_E)$). Because quantifications were performed from different portions ($n \geq 10$) of the dataset, average and standard deviations were calculated to represent quantifications for the dataset (control or TOF hearts, regions A and B).

Myofibril orientation (angle) quantifications were also performed. To this end, we used the myofibril segmentations in the transmural image plane (x-y), split them into individual segmentations (each myofibril was separately assessed using the multi-ROI functions in Dragonfly) and quantified the transmural angle of each myofibril, as the angle with respect to the x-direction, θ angle. Because the x-direction was approximately parallel to the wall boundary, θ = 0° represents myofibrils that are oriented parallel to the wall or in circumferential direction, and θ = 90° are myofibrils oriented towards the ventricle lumen in the radial direction. Angular quantifications were plotted as histograms showing frequency (number of myofibrils) versus an angular range. For quantifications in the x-z plane (a plane approximately parallel to the heart wall), we re-sampled the images acquired in the x-y plane, to obtain images in the x-z plane, and then performed the quantifications as before. We quantified the elliptical orientation angle of each myofibril (x-z plane) with respect to the x-direction, Φ angle. Thus Φ = 0°, represents myofibrils that are oriented circumferentially around the ventricle, and θ = 90° are myofibrils oriented longitudinally along the ventricular wall (see *Figure 10*). Results were plotted as histograms, from which elliptical and transmural angular orientations were further characterized using average and standard deviation data.

## Acknowledgements

This work has been funded by a grant from US National Institutes of Health, NIH R01 HL094570 (SR); the OHSU University Shared Resource pilot funding program (SR); the OHSU School of Medicine Faculty Innovation Fund (SR). We thank Kevin Loftis for graciously sharing his time and knowledge of Amira. We also thank Melissa Williams for her help in optimizing our sample preparation and Renovo Neural, Inc, especially Emily Benson and Grahame Kidd, for sharing their SBF-SEM protocol. Electron microscopy was performed at the OHSU Multiscale Microscopy Core (MMC) with technical support from the OHSU Center for Spatial Systems Biomedicine (OCSSB).

An extended abstract (two pages) of this work was published and presented at Microscopy & Microanalysis 2019 Meeting in Portland, OR, https://www.cambridge.org/core/journals/microscopy-and-microanalysis/article/multiscale-cardiac-imaging-from-whole-heart-images-to-cardiac-ultrastructure/7053ED929882C2E43B2ED85FD6D78BEC.

## Additional information

### Funding

| Funder | Grant reference number | Author |
|---|---|---|
| National Institutes of Health | R01 HL094570 | Sandra Rugonyi |
| Oregon Health and Science University | | Sandra Rugonyi |
| School of Medicine, Oregon Health and Science University | | Sandra Rugonyi |

The funders had no role in study design, data collection and interpretation, or the decision to submit the work for publication.

### Author contributions

Graham Rykiel, Investigation, Writing - review and editing, Prepared the samples, acquired micro-CT images, and analyzed micro-CT and EM images; Claudia S López, Conceptualization, Formal analysis, Validation, Methodology, Writing - review and editing, Optimized the sample preparation protocols and acquired the EM images; Jessica L Riesterer, Validation, Investigation, Methodology, Writing - review and editing, Optimized the sample preparation protocols and acquired the EM images; Ian Fries, Data curation, Formal analysis, Visualization, Segmented and quantified the SBF-SEM images; Sanika Deosthali, Data curation, Visualization, Segmented micro-CT images for comparison; Katherine Courchaine, Visualization, Writing - review and editing, Analyzed images and

assisted GR with figures; Alina Maloyan, Formal analysis, Validation, Writing - review and editing, Analyzed the quality of images and helped with biological image interpretation; Kent Thornburg, Formal analysis, Writing - review and editing, analyzed the quality of images and helped with biological image interpretation; Sandra Rugonyi, Conceptualization, Resources, Formal analysis, Supervision, Funding acquisition, Writing - original draft, Writing - review and editing

## Author ORCIDs
Jessica L Riesterer (iD) https://orcid.org/0000-0003-1084-2773
Alina Maloyan (iD) https://orcid.org/0000-0002-7309-5026
Sandra Rugonyi (iD) https://orcid.org/0000-0001-9262-7959

## Decision letter and Author response
Decision letter https://doi.org/10.7554/eLife.58138.sa1
Author response https://doi.org/10.7554/eLife.58138.sa2

# Additional files

## Supplementary files
• Transparent reporting form

## Data availability
Data generated and analysed during this study are described in the manuscript. Source data files have been provided for Figures 8 and 10. Datasets have been submitted to Dryad with DOI https://doi.org/10.5061/dryad.hdr7sqvg5.

The following dataset was generated:

| Author(s) | Year | Dataset title | Dataset URL | Database and Identifier |
|---|---|---|---|---|
| Rugonyi S | 2020 | Multiscale Heart Image Data | https://doi.org/10.5061/dryad.hdr7sqvg5 | Dryad Digital Repository, 10.5061/dryad.hdr7sqvg5 |

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
