## [Decision Letter]

**Acceptance summary:**

Your proof-of-principle article provides a very useful multi-scale approach to correlate ultrastructural and cellular findings with tissue geometry and structure relevant to cardiac morphogenesis and malformations. This general approach can be applied to a wide range of questions relevant to developmental biology and the origins of congenital malformations.

**Decision letter after peer review:**

Thank you for submitting your work entitled "Multiscale imaging of the whole heart and its internal cellular architecture: Application to congenital heart disease" for consideration by *eLife*. Your article has been reviewed by three peer reviewers, one of whom is a member of our Board of Reviewing Editors, and the evaluation has been overseen by a Senior Editor. The following individual involved in review of your submission has agreed to reveal their identity: Karthik Kodigepalli (Reviewer #3).

Our decision has been reached after consultation between the reviewers. Based on these discussions and the individual reviews below, we regret to inform you that your work will not be considered further for publication in *eLife*.

The reviewers found the current manuscript to present an interesting approach of combining micro-CT imaging with regional histology in the avian heart, however, 2 of the 3 reviewers felt that the findings of the current manuscript are too preliminary to warrant publication in *eLife* at this time. In addition to text edits, 2 of the 3 reviewers recommend a larger number of embryos are required to determine the accuracy of the proposed methods, including sufficient numbers for statistical analysis. The manuscript would also be strengthened by quantitative alignment analysis in addition to qualitative comments.

Reviewer #1:

Essential revisions:

1) The current title suggests that the described methods are applicable to a better understanding of congenital heart disease (CHD). However, it would be more accurate to state the current described findings in 2 non-beating chick hearts (1 normal, 1 with VSD and conotruncal abnormalities) is a "Proof of Principle" for correlating changes in macroscale abnormalities in cardiac structure with microscale changes in tissue and cell composition (as mentioned in the Introduction).

2) The authors should comment on, aside from the technical accomplishments of the approach, how this data provides new insights into errors in cardiac morphogenesis, altered growth and adaptation, or the changes in mechanical function that occur in CHD.

3) The authors should discuss, briefly, how changes in myofiber architecture are related to mechanical function (chamber compliance and contractility,.…) since changes in mechanical loading and function can result in multiscale changes in cardiac architecture.

4) The paper would be strengthened by the analysis of larger number of imaged hearts (normal and abnormal) to better understand the reproducibility of the described methods. This paper uses a single normal and a single abnormal heart, and describes the results from 2 regions of interest from the LV free wall of each specimen. The only significant difference between the 2 specimens was the increase in extracellular space noted in the TOF specimen (Figure 7B). The paper would be strengthened, for example, by the analysis of adjacent ROIs in the same heart to determine reproducibility and by comparison of specimens from both LV and RV specimens.

5) The authors highlight the ability to classify phenotype, heart size, and wall thickness with their technique (subsection “Comparison of control and TOF hearts and limitations of this study”), however the paper lacks any discussion of the now readily available and less time-consuming approach of using of high-resolution echocardiography to acquire data on multiple time points from beating avian and mammalian developing hearts. It is important to explain how the multiscale data derived from this approach complements currently available data sets.

6) It is important for the authors to reference the studies that detailed the congenital cardiac anomalies associated with conotruncal banding in the chick embryo which are associated with abnormal neural crest migration, altered aortic-mitral valve continuity, aortic valve override, VSDs, and abnormal conotruncal septation (Gessner, 1966 and Clark et al., 1984). This mechanism for altered conotruncal septation may be very different from the altered neural crest and conotruncal anomalies in the chick embryo produced by neural crest ablation (Hutson and Kirby, 2003) which may impact the changes in myocardial and myofiber orientation and maturation.

The technical details of the current studies are excellent (with the exception of the lack of quantitative data on cellular orientation).

Introduction:

This sentence can be more clearly restructured. It would be more accurate to state that most studies of the myocardial architecture, including the analysis of myofiber architecture and cellular constituents have found all to be abnormal in the setting of congenital cardiac malformations. What is unknown is the mechanisms responsible and the subsequent clinical consequence…

It would not be accurate to state that the results of a small number of studies on myofiber architecture have been "ignored while planning treatment strategies" and would probably be more accurate to say that those myofiber studies have not been proven to be relevant to decisions regarding treatment strategies. This sentence should simply state that emerging evidence supports the importance of myofiber disarray in CHD which may impact cardiac function before and after interventional procedures.

The authors should include 1 or 2 references to other imaging approaches, such as OCT and the use of high-resolution echocardiography for cardiovascular phenotyping of CHD in developing avian and mammalian developing hearts.

The authors should provide an updated pathogenesis that TOF results from abnormal neural crest cell migration and altered aortopulmonary septation which can present with a range of phenotypes including four basic features (i, ii, iii, and iv…) (Hutson and Kirby, 2003).

Figure 4. The RV and LV should be identified in the CON and TOF hearts.

Subsection “Image segmentation and quantification” and Figure 8. The authors suggest that Figure 8 shows a change in the orientation of myocardial cells but no quantitative data (or data on specimen reproducibility) is provided. This impression would be strengthened by some quantitative data and some evidence that a known external reference point was used to orient the specimens prior to histologic processing.

Subsection “Conclusions” The text suggests that this approach "enables analysis of both whole heart and ultrastructural architecture" however only a small segment of the ultrastructure was assessed. It is more accurate to say, "The described approach allows the correlation of macro and micro-scale architecture in selected regions of the heart".

Reviewer #2:

The manuscript by Rykiel reports an improved method for microCT analysis of heart anatomy followed by ultrastructural analysis of myocardial cells in the same sample. Analysis of a control and malformed embryonic chicken heart is shown as proof of principal for the utility of this technique. The main advantage of the reported method is improved fixation and imaging of the same samples. However, there are several limitations of the method as shown in the current study.

Essential revisions:

1) In order for this method to be useful in rigorous analysis of cardiac anatomy and cellular organization, additional quantification and evidence for reproducibility of the methods are needed. The current manuscript analyzing n=1 sample sizes is inadequate to determine if reproducible quantitative data can be obtained using these methods.

2) Quantification of additional parameters in the microCT and SEM studies are needed.

3) Tests of statistical significance were performed based on multiple measurements of the same samples. These tests are not appropriate for the n=1 sample sizes in the current study.

4) It is difficult to appreciate the benefit of the 3D ultrastructural analysis from the images shown. It is difficult to appreciate the features of interest in Figure 6, Figure 7 and Figure 8. What are the red and green stains showing in Figure 8?

5) The main conclusion related to differences in the control and TOF heart seems to be at there is increased extracellular space related to increased trabeculation. This likely would be easier to appreciate on a lower resolution analysis or ventricular myocardial architecture.

6) The authors emphasize that this method could be applied to analysis of human congenital heart disease. The reported sample preparation and ultrastructural analysis is not feasible at a whole organ level for humans.

7) This protocol might be useful for analysis of cardiac malformations in mice. Are there any data to support the use of this method in other animal models?

Reviewer #3:

This manuscript titled "Multiscale imaging of the whole heart and its internal cellular architecture: Application to congenital heart disease" by Rykiel et al., presents a multiscale and correlative imaging system that combines the 3D-micro-computed tomography and scanning electron microscopy techniques to simultaneously assess micro and ultrastructure of whole heart tissues. The authors have utilized an avian (chick) in vivo model and have satisfactorily demonstrated the applications of their novel imaging system. Micro-CT and SEM techniques have been individually well established and combining both can be challenging due different ensuing image resolutions and problems with fixation and staining. In the avian heart model, this study indicates to have overcome these problems and successfully applied this technique to larger tissues (5-6 mm wide) than the previous studies. This novel method can indeed be a powerful tool in studying the CHD and other disease models. The procedures and steps of optimized staining, fixation and imaging using micro-CT and SBF-SEM are clearly illustrated and described in good detail. There are, however, some minor issues that need to be addressed in the manuscript to improve the clarity and before the study can be published.

Comments:

1) Subsection “Cardiac structure analysis from 3D micro-CT images”, while analyzing the micro-CT images of control avian heart vs the TOF avian heart, authors mention that in the TOF heart, 'right' branch of the pulmonary artery was missing compared to control heart. In support of this observation, they refer to a rare human condition 'unilateral absence of pulmonary artery' seen in TOF patients. However, in Figure 3 legend, they mention that the 'left' branch of pulmonary artery was absent in the TOF heart. Was this observation made in the other sample? It is confusing to the reader and needs to be clarified.

2) Article's main focus in on the multiscale imaging technique and its potential applications. The data presented in Figure 7 and Figure 8 are obtained from one control and one TOF avian heart. Although authors address this limitation in the article, one would need to be careful not to conclude much based on the observations comparing CT and TOF hearts although they are in line with previously published findings.

3) In subsection “Image segmentation and quantification”, authors mention while describing the 3D visualization of SBF-SEM segmentations (Figure 8), that orientation of myocardial cells were 'slightly' different. This needs to be described further as to how exactly the orientation of TOF myocardial cells was different from the control.

4) Other potential applications of this novel imaging technique in addition to CHDs (extracellular matrix disorganization, fibrosis, myxomatous degeneration of valve tissues etc.) need to be briefly commented on. Any other cardiac ultra-structures that can potentially be studied using this technique must be mentioned briefly.

5) The novel imaging technique presented here improves on the previous methods by being applicable to larger tissues (up to 5-6mm according to authors). However, this is still smaller tissue size compared to other in vivo models and human heart tissues. Do authors foresee any potential problems in translating this technology to human heart tissues? This needs to be elaborated upon.

---

## [Author Response]

Reviewer #1:Essential revisions:1) The current title suggests that the described methods are applicable to a better understanding of congenital heart disease (CHD). However, it would be more accurate to state the current described findings in 2 non-beating chick hearts (1 normal, 1 with VSD and conotruncal abnormalities) is a "Proof of Principle" for correlating changes in macroscale abnormalities in cardiac structure with microscale changes in tissue and cell composition (as mentioned in the Introduction).

Thanks for the comment. Our current title “Multiscale imaging of the whole heart and its internal cellular architecture: Application to congenital heart disease” certainly suggests that we have analyzed congenital heart disease extensively, which is not the case. Our intention was to highlight this possibility. We agree to change the title to “Multiscale cardiac imaging spanning the whole heart and its internal cellular architecture in a small animal model” to more accurately reflect the paper scope of presenting the multiscale imaging procedure, with examples supporting its use.

2) The authors should comment on, aside from the technical accomplishments of the approach, how this data provides new insights into errors in cardiac morphogenesis, altered growth and adaptation, or the changes in mechanical function that occur in CHD.

Thanks for making this point. We have added a paragraph explaining this in more detail. Briefly, the imaging method presented will allow us to determine how the heart adapts to hemodynamic conditions (the case presented) but also to other insults (e.g. hyperglycemia, hypoxia), and draw comparisons among different cases and normal hearts. The combination of structural and ultrastructural cardiac imaging data, moreover, can predict mechanical function. Indeed, myocardial disarray or marked reductions in myofibrils, for example, will decrease cardiac efficiency, including compromised ejection fraction and stroke volume.

We added the following paragraph in the Discussion section of the paper:

“The presented correlative multiscale imaging approach will provide new insights into the underpinnings of heart development, and on cardiac mechanical changes in CHD. Analysis of multiscale cardiac images over developmental stages, will unravel how cardiac tissues progressively assemble and mature at diverse scales to form a functional heart. When cardiac development is perturbed (due to altered blood flow, hyperglycemia, hypoxia or any teratogen substance or changes in the environment) multiscale images will reveal how the natural maturation and assembly of cardiac components (cells, extracellular matrix) are altered, sometimes in irreversible ways that lead to CHD. Multiscale images, furthermore, promise to reveal possible differences and similarities among CHD hearts with different phenotypes (e.g. TOF vs DORV), as well as among hearts with similar phenotypes but that were the result of diverse insults (e.g. TOF hearts that arise due to altered blood flow vs TOF hearts due to deficiencies in the Hedgehog gene). Moreover, the combination of morphological and ultrastructural cardiac imaging data can predict the mechanical function of the heart. Indeed, myocardial disarray or marked reductions in myofibrils, for example, will decrease the contraction efficiency (and force) of cardiac tissues. In turn, decrease contractility affects cardiac efficiency, including compromised ejection fraction and stroke volume.”

3) The authors should discuss, briefly, how changes in myofiber architecture are related to mechanical function (chamber compliance and contractility,.…) since changes in mechanical loading and function can result in multiscale changes in cardiac architecture.

Thanks for this insightful suggestion. We agree that this is missing, and we have added to this, explaining how myocardial cell architecture impacts cardiac function. Briefly, myofibers contract along their long axes and align parallel to each other, so that cardiac contraction occurs along specific directions in the heart. The normal helical architecture of myofibers ensures that the heart contracts and twists to effectively eject blood into the pulmonary and systemic circulations. If the myofiber architecture is perturbed, however, cardiac contractility is compromised. This is both because the tissue can no longer contract on a specific direction, and because contraction force may be diminished. Thus, changes in myofiber architecture (with respect to normal) frequently lead to heart tissues with impaired contractility and thus reduced compliance.

We added the following sentences in the Introduction:

“This highly organized pattern ensures that cardiac contraction occurs along specific directions, so that the heart can effectively eject blood into the pulmonary and systemic circulations. Perturbations of the normal myocardial architecture affect cardiac contractility and compliance. This is both because the tissue can no longer contract on the very specific directions and patterns that optimize cardiac function, but also because cardiac contraction force may be diminished. Therefore, changes in myocardial architecture (with respect to normal) frequently lead to heart tissues with impaired contractility and reduced compliance.”

4) The paper would be strengthened by the analysis of larger number of imaged hearts (normal and abnormal) to better understand the reproducibility of the described methods. This paper uses a single normal and a single abnormal heart, and describes the results from 2 regions of interest from the LV free wall of each specimen. The only significant difference between the 2 specimens was the increase in extracellular space noted in the TOF specimen (Figure 7B). The paper would be strengthened, for example, by the analysis of adjacent ROIs in the same heart to determine reproducibility and by comparison of specimens from both LV and RV specimens.

We have developed the methods described after several iterations that optimized tissue fixation and preparation for both the microCT and SEM images. Once we found the combination of procedures to achieve the multiscale resolution presented in this paper, we chose two hearts (the normal and abnormal heart specimens). We applied the methods to those two hearts to show that we could successfully reproduce proper fixation and preparation of tissues required for high resolution imaging at diverse scales, even though the hearts were different. In other words, we showed that the method works not only for normal hearts, but also for malformed hearts. Indeed, the methods worked without additional optimizations or tweaks. Adding more hearts to the study will be useful for a Research paper, with the objective of understanding how abnormal hearts differ among themselves and with respect to normal; and how perturbed blood flow conditions or other insults lead to different adaptations and ultrastructure. We therefore respectfully disagree with the reviewer: we do not think we need more samples to show, in a Tools and Resources article, that our method works. Instead, more samples will be required for a full Research paper focused on comparisons of hearts and insights gained by the methods and analyses of images.

We agree that the paper will be strengthened by analysis of adjacent ROIs from the same hearts, which we have performed. By analyzing SEM images that were 2 microns apart (results presented in Figure 7A – now Figure 8A) we accomplished adjacent ROI analysis and comparison, and further found that differences among these adjacent ROIs were small (not more than 10%), emphasizing the reproducibility of our imaging and segmentation methods. Including the RV as well as the LV would be nice – however, the main objective of the analysis presented was not to do an extensive comparison of these two hearts, but show possible types of analyses, and the kind of insights that can be gained. The fact that the only difference we could find in the LV of these two hearts is increased extracellular space is not important (and not significant), the point we want to emphasize is that our approach enables cardiac multiscale imaging and quantitative analysis. A more extensive study with more hearts (a full Research paper) could then show similarities and differences among hearts at multiple scales, as well as perhaps unique adaptations in diverse regions of the same heart.

5) The authors highlight the ability to classify phenotype, heart size, and wall thickness with their technique (subsection “Comparison of control and TOF hearts and limitations of this study”), however the paper lacks any discussion of the now readily available and less time-consuming approach of using of high-resolution echocardiography to acquire data on multiple time points from beating avian and mammalian developing hearts. It is important to explain how the multiscale data derived from this approach complements currently available data sets.

Thanks for bringing this important point to our attention. We are well aware of the currently available echocardiography imaging (we have published a study that analyzes both normal and malformed hearts using a combination of echocardiography and microCT [Midgett, Thornburg and Rugonyi, 2017]). Using echocardiography we can certainly obtain data on heart function from beating avian and mammalian hearts. Echocardiography data can be further used to quantify heart size and wall thickness, but also blood flow velocities, ejection fractions and other functional cardiac parameters. However, while we can observe functional problems in hearts using echocardiography, we cannot get the ultrastructural details of the hearts. Moreover, it is also difficult to identify, using only echocardiography, subtle malformations in the heart structure (this is the reason we used a combination of echocardiography and microCT in our study, to phenotype hearts more precisely). Our multiscale approach, therefore, can certainly complement echocardiography data: by providing a more detailed phenotyping of the hearts, including structural and ultrastructural characteristics that can explain and eventually predict in vivo function.

We have now addressed these considerations in the Introduction: “For later stages of heart development ecochardiography is used in mice and chick. In small animal research, moreover, due to the small size of developing hearts, functional imaging techniques are frequently complemented with micro-CT or histology to more accurately phenotype the heart.”

And in the Discussion section: “…combining echocardiography, which can acquire in vivo images of the heart for functional (including blood flow) analysis, together with 3D micro-CT and SEM, can reveal functional as well as detailed microstructural and ultrastructural characteristics of normal versus CHD hearts.”

–

6) It is important for the authors to reference the studies that detailed the congenital cardiac anomalies associated with conotruncal banding in the chick embryo which are associated with abnormal neural crest migration, altered aortic-mitral valve continuity, aortic valve override, VSDs, and abnormal conotruncal septation (Gessner, 1966 and Clark et al., 1984). This mechanism for altered conotruncal septation may be very different from the altered neural crest and conotruncal anomalies in the chick embryo produced by neural crest ablation (Hutson and Kirby, 2003) which may impact the changes in myocardial and myofiber orientation and maturation.

This is indeed an important point for discussion and consideration. Thanks for highlighting this point. We have added a short explanation clarifying that conotruncal heart defects, similar to the ones obtained after hemodynamic interventions, can also result from neural crest ablation, adding the references mentioned by the reviewer. Our multiscale imaging method could then be useful to determine possible changes in ultrastructure and adaptation in a seemingly similar cardiac malformation phenotype but due to diverse origins.

We have added the following paragraph in the Discussion section:

“Previous studies have detailed the congenital heart anomalies associated with outflow tract banding (OTB) used in this study to induce TOF. Those studies found a spectrum of congenital heart defects after OTB, which originated from abnormalities in the outflow tract (conotruncal defects). These abnormalities included increased separation between the aortic and mitral valve annuli (altered aortic-mitral valve continuity), ventricular septal defects, abnormal position of the aorta, including TOF and double outlet right ventricle (DORV), in which both the aorta and pulmonary trunks emerge from the right ventricle. Meanwhile, neural crest cell ablation also leads to conotruncal anomalies in the chick embryo. Cardiac neural crest cells are required for normal heart development (in the chick and mouse), and ablation of these cells leads to persistent truncus arteriosus (PTA), characterized by lack of separation of the aorta and pulmonary trunk, but also to TOF and DORV. Likewise, diverse genetic anomalies are also associated with conotruncal heart defects. However, the mechanisms by which anomalous genes, neural crest cell ablation, and altered hemodynamics lead to conotruncal defects may differ, and these differences may impact myocardial and myofiber orientation and maturation. Multiscale imaging studies could reveal the impact of diverse interventions on cardiac microstructure and ultrastructure, both when the same or different phenotypes are obtained. This in turn could contribute to our understanding of the underpinnings of CHD and their functional and structural consequences.”

The technical details of the current studies are excellent (with the exception of the lack of quantitative data on cellular orientation).

We apologize for this omission. We have added the quantification of myofibrils cell orientation from our images in the Results section and added Figure 10 to illustrate it.

Introduction:This sentence can be more clearly restructured. It would be more accurate to state that most studies of the myocardial architecture, including the analysis of myofiber architecture and cellular constituents have found all to be abnormal in the setting of congenital cardiac malformations. What is unknown is the mechanisms responsible and the subsequent clinical consequence…

The reviewer is correct. We have rewritten this sentence as follows: “The structural (morphological or “geometrical”) characteristics of heart malformations, including changes in cardiac wall architecture, have been extensively studied. However, the mechanisms that lead to an anomalous cardiac architecture in CHD and the clinical consequences of it are unknown. “

It would not be accurate to state that the results of a small number of studies on myofiber architecture have been "ignored while planning treatment strategies" and would probably be more accurate to say that those myofiber studies have not been proven to be relevant to decisions regarding treatment strategies. This sentence should simply state that emerging evidence supports the importance of myofiber disarray in CHD which may impact cardiac function before and after interventional procedures.

Yes, we agree with the reviewer. We have edited this sentence, which now states “These emergent studies are revealing myocardial disarray in CHD (with respect to their normal counterparts) that very likely affect cardiac function before and after surgical repair.”

The authors should include 1 or 2 references to other imaging approaches, such as OCT and the use of high-resolution echocardiography for cardiovascular phenotyping of CHD in developing avian and mammalian developing hearts.

Thanks for pointing this out. We have included references to other methodologies employed for phenotyping. We are aware of these techniques and have used them in our work. OCT is used for earlier stages, when the heart is still small and tubular (before about 4 days of incubation in chick). For the studies presented here, when the heart already has four chambers (day 12 of incubation in chick) OCT, an optical imaging technique, cannot penetrate the tissue, and instead high-resolution echocardiography is used in avian and mammalian developing hearts. In our own studies [Midgett, Thornburg and Rugonyi, 2017], further, for the chick hearts echocardiography was used for phenotyping but was complemented by microCT, from which we could get a more detailed image of the heart morphology.

We have added a paragraph on the use of OCT and echocardiography techniques in the context of heart development and CHD: “Imaging approaches have been employed to study cardiac function in normal and CHD hearts. 8 In humans, ultrasound-based echocardiography, which can image the in vivo motion of cardiac walls during heart pumping and measure blood flow velocities within the heart, is used to diagnose fetal CHD in utero and assess severity. After the baby is born, echocardiography is useful to monitor cardiac function before and after interventions to repair CHD. Magnetic resonance imaging (MRI) is also employed to precisely diagnose malformations and monitor cardiac function in CHD. In heart development animal research, moreover, optical coherence tomography (OCT) and echocardiography are used with avian and mammalian models of CHD. OCT, like ultrasound, is a non-invasive technique that can image heart motion and measure blood flow velocities within the heart. OCT resolution (< 5 µm) is ideal for early avian and mouse embryos during tubular-heart developmental stages. For later stages of heart development ecochardiography is used in mice and chick. In small animal research, moreover, due to the small size of developing hearts, functional imaging techniques are frequently complemented with micro-CT or histology to more accurately phenotype the heart.”

–

The authors should provide an updated pathogenesis that TOF results from abnormal neural crest cell migration and altered aortopulmonary septation which can present with a range of phenotypes including four basic features (i, ii, iii, and iv…) (Hutson and Kirby, 2003).

We would like to thank the reviewer for bringing this important point out. TOF can certainly result from other mechanisms, including abnormal neural crest cell migration, which is extremely interesting. Originally, we had just focused the presentation and discussion of the paper on the model we were presenting (hemodynamic interventions), but it is fair to also mention other possible causes of TOF. Interestingly, our proposed multiscale imaging method can be used to compare hearts with TOF from perturbations in hemodynamics versus perturbations in neural crest cell migration. We have added in the Discussion section that TOF can result from abnormal neural crest cell migration as well as provide the reference cited by the reviewer.

We have added the following paragraph to the Discussion section:

“Previous studies have detailed the congenital heart anomalies associated with outflow tract banding (OTB) used in this study to induce TOF. Those studies found a spectrum of congenital heart defects after OTB, which originated from abnormalities in the outflow tract (conotruncal defects). These abnormalities included increased separation between the aortic and mitral valve annuli (altered aortic-mitral valve continuity), ventricular septal defects, abnormal position of the aorta, including TOF and double outlet right ventricle (DORV), in which both the aorta and pulmonary trunks emerge from the right ventricle. Meanwhile, neural crest cell ablation also leads to conotruncal anomalies in the chick embryo. Cardiac neural crest cells are required for normal heart development (in the chick and mouse), and ablation of these cells leads to persistent truncus arteriosus (PTA), characterized by lack of separation of the aorta and pulmonary trunk, but also to TOF and DORV. Likewise, diverse genetic anomalies are also associated with conotruncal heart defects. However, the mechanisms by which anomalous genes, neural crest cell ablation, and altered hemodynamics lead to conotruncal defects may differ, and these differences may impact myocardial and myofiber orientation and maturation. Multiscale imaging studies could reveal the impact of diverse interventions on cardiac microstructure and ultrastructure, both when the same or different phenotypes are obtained. This in turn could contribute to our understanding of the underpinnings of CHD and their functional and structural consequences.”

Figure 4. The RV and LV should be identified in the CON and TOF hearts.

Thanks for pointing to this omission. We have re-label Figure 4, in particular labels for the right and left ventricles (RV and LV, respectively) have been added.

Subsection “Image segmentation and quantification” and Figure 8. The authors suggest that Figure 8 shows a change in the orientation of myocardial cells but no quantitative data (or data on specimen reproducibility) is provided. This impression would be strengthened by some quantitative data and some evidence that a known external reference point was used to orient the specimens prior to histologic processing.

We agree with the reviewer (please note that Figure 8 is now Figure 9).

We have now provided the myofibril angle quantifications, which are summarized in Figure 10 and described in the text of the manuscript (both in the Results section and the Materials and methods section).

Subsection “Conclusions” The text suggests that this approach "enables analysis of both whole heart and ultrastructural architecture" however only a small segment of the ultrastructure was assessed. It is more accurate to say, "The described approach allows the correlation of macro and micro-scale architecture in selected regions of the heart".

We agree with the reviewer and have amended the sentence accordingly: “The described approach allows the correlation of microstructural and ultrastructural architecture in selected regions of the heart.”

Reviewer #2:The manuscript by Rykiel reports an improved method for microCT analysis of heart anatomy followed by ultrastructural analysis of myocardial cells in the same sample. Analysis of a control and malformed embryonic chicken heart is shown as proof of principal for the utility of this technique. The main advantage of the reported method is improved fixation and imaging of the same samples. However, there are several limitations of the method as shown in the current study.

Thanks for your positive view of the manuscript and insightful comments. We will address study limitations pointed by the reviewer in what follows.

Essential revisions:1) In order for this method to be useful in rigorous analysis of cardiac anatomy and cellular organization, additional quantification and evidence for reproducibility of the methods are needed. The current manuscript analyzing n=1 sample sizes is inadequate to determine if reproducible quantitative data can be obtained using these methods.

We respectfully disagree with the reviewer on this. We do not think we need more samples to show evidence of reproducibility of the methods. We have developed the methods described after several iterations that optimized tissue fixation and preparation for both the microCT and SEM images. Once we found the combination of procedures that achieved multiscale resolution as presented in this paper (described in details in the Materials and methods section), we chose two hearts (the normal and abnormal hearts presented) and applied the methods to those two hearts. We showed that we could nicely replicate the methods (proper fixation and preparation of tissues for imaging), even though the hearts were different (in other words, that the method would not only work for normal hearts, but also abnormal ones). The method worked seamlessly (we did not need additional optimizations or tweaks) and we could get homogeneous fixation and staining in both hearts (as presented in the manuscript) as well as high-resolution images of heart structure and ultrastructure.

We agree that adding more hearts to the study will be useful for a Research paper, in which the aim is to understand how similar/different abnormal hearts are among themselves and with respect to normal. Such paper would indeed focus on comparisons of hearts and insights gained by application of the imaging approach presented. We do not think, however, that more samples/animals are needed for a Tools and Resources paper (the article type for this manuscript), in which the focus of the paper is on the methods and procedures needed to achieve multiscale imaging of a relatively large organ, the heart.

2) Quantification of additional parameters in the microCT and SEM studies are needed.

We have added more quantifications from microCT and 3D SEM images in the Results. These include quantifications of ventricle sizes from microCT and myofibril alignment from SEM. We initially chose to not include too many quantifications not to mislead the readers. Given that n = 1 (1 normal heart, and 1 TOF heart), we cannot make conclusions from a biological point of view, and we did not want to give the wrong impression in this Tools and Resources paper. However, more quantifications have now been included to illustrate additional data that can be obtained from our presented multiscale imaging approach, while warning the reader that data is for illustration purposes only.

3) Tests of statistical significance were performed based on multiple measurements of the same samples. These tests are not appropriate for the n=1 sample sizes in the current study.

Statistical tests on multiple measurements from the same samples (different locations) were only included to show whether the two samples were different. We have removed these tests not to confuse readers (statistical tests are typically performed when comparing animal groups), and also because they are not necessary (differences are obvious without the statistical tests).

4) It is difficult to appreciate the benefit of the 3D ultrastructural analysis from the images shown. It is difficult to appreciate the features of interest in Figure 6, Figure 7 and Figure 8. What are the red and green stains showing in Figure 8?

Thanks for the comment. We apologize for not showing more clearly the advantages of 3D SEM from our figures. We have added a Figure (Figure 7) showing the detailed architecture obtained from SEM in a small region, that illustrates the advantages of 3D SEM.

Regarding the specific figures mentioned by the reviewer. Figure 6 just shows that ultrastructural details can be appreciated on the imaging plane (x-y plane) but also on perpendicular planes (x-z and y-z planes), and that the images acquired (part of the image stack) are well aligned and homogeneous. We have added labels to the figure to make it more clear, and edited its caption. Figure 7 (now Figure 8) simply shows quantifications based on SEM images in A and B, and then a depiction of the 3D extracellular space in C and D for the control and TOF hearts, respectively. We have edited the caption to more accurately reflect this. Figure 8 (now Figure 9) shows a completely segmented 3D SEM volume. Please note that the ‘stains’ are segmentations (painted delineations) of the nuclei (red), mitochondria (blue) and myofibrils (green). These segmentations were obtained semi-automatically using a deep learning (artificial intelligence) strategy. We are very sorry we omitted to mention in the figure caption what the colors represent, and we have now added this explanation to the figure caption. In addition, as mentioned before, we have added a figure (Figure 7) showing detailed segmentations of ultrastructure.

5) The main conclusion related to differences in the control and TOF heart seems to be at there is increased extracellular space related to increased trabeculation. This likely would be easier to appreciate on a lower resolution analysis or ventricular myocardial architecture.

We agree with the reviewer: increased trabeculation could be better appreciated with other methods. However, we cannot conclude anything of biological significance from n = 1. We apologize for not being clear on this, and we have clarified this in the paper now. The analysis performed on images was not exhaustive, and was only presented as an example of possible quantifications. There are other aspects that were not fully analyzed, like lipid droplet distribution, glycogen distribution, the distribution of the mitochondria with respect to the myofibrils, etc. Our intention on selecting nuclei, mitochondria and myofibrils for segmentation, and to perform an initial quantification of them, was to show possible quantitative outcomes, and merely as an example. More detailed quantifications could be performed – and would indeed be appropriate for a paper analyzing differences between normal or control hearts and CHD hearts, in which also appropriate number of animals, and careful selection of SEM ROIs including both the left and right ventricles, are considered. The fact that for these two hearts there was no difference in myocardial cell volume fraction of nuclei, myofibrils and mitochondria, in their left ventricles was coincidental. Thus, while it seems that the main conclusion from the analysis is that left ventricle trabeculation and the extracellular space is increased in TOF, and that therefore other techniques are more suitable (and cheaper) to analyze these two hearts, this conclusion is misleading, as there is more information from 3D SBF-SEM images that we have not analyzed that cannot be obtained from low resolution images of the ventricular myocardial architecture. Further, this conclusion could pertain exclusively to these two hearts, with other hearts perhaps showing differences.

We have added the following in the Discussion section: “To be biologically meaningful, however, these studies need to include more animals. The quantifications and comparisons presented here for one control and one TOF heart (thus n=1) pertain only to these two hearts, and are presented as an illustration of possible ways of extracting information from the proposed multiscale imaging method. “

6) The authors emphasize that this method could be applied to analysis of human congenital heart disease. The reported sample preparation and ultrastructural analysis is not feasible at a whole organ level for humans.

We agree. Sorry the text was misleading. The method is not intended for human hearts, it is intended for small animal models of congenital heart disease, which could inform human congenital heart disease. We have now clarified this more emphatically in the manuscript.

We have added the following paragraph to the Discussion section:

“Direct application of the proposed multiscale imaging method to human hearts is limited. The method presented here is destructive, and thus can only be applied to human samples of diseased individuals. Moreover, the larger size of the human heart will lead to difficulties in attaining homogeneous fixation and staining. To circumvent fixation and stain homogeneity issues, increased timing for diffusion of fixative and heavy metal stains is certainly a possibility as is microwave steps (to accelerate diffusion). In addition, changes in processing, such as sectioning of hearts after microCT to facilitate diffusion of heavy metal stains and control imaging are also possible. As presented, using our methods, multiscale imaging of human hearts is limited.”

7) This protocol might be useful for analysis of cardiac malformations in mice. Are there any data to support the use of this method in other animal models?

Thanks for bringing this important point. In fact, mice hearts are very similar in size to chicken hearts, and we mentioned in the manuscript that we could apply the same techniques to mice. Unfortunately, while we have some microCT images of fetal mice hearts, and others have acquired 2D EM images of mouse hearts, we have not yet performed a full multiscale image as described in the manuscript on a mouse heart. However, we anticipate that the techniques will translate seamlessly to mice hearts with perhaps only a few (if any) small optimizations.

Reviewer #3:This manuscript titled "Multiscale imaging of the whole heart and its internal cellular architecture: Application to congenital heart disease" by Rykiel et al., presents a multiscale and correlative imaging system that combines the 3D-micro-computed tomography and scanning electron microscopy techniques to simultaneously assess micro and ultrastructure of whole heart tissues. The authors have utilized an avian (chick) in vivo model and have satisfactorily demonstrated the applications of their novel imaging system. Micro-CT and SEM techniques have been individually well established and combining both can be challenging due different ensuing image resolutions and problems with fixation and staining. In the avian heart model, this study indicates to have overcome these problems and successfully applied this technique to larger tissues (5-6 mm wide) than the previous studies. This novel method can indeed be a powerful tool in studying the CHD and other disease models. The procedures and steps of optimized staining, fixation and imaging using micro-CT and SBF-SEM are clearly illustrated and described in good detail. There are, however, some minor issues that need to be addressed in the manuscript to improve the clarity and before the study can be published.

We greatly appreciate your positive view of our work, and we would like to thank you for taking the time to review our manuscript. We have addressed issues in the manuscript as detailed below.

Essential revisions:1) Subsection “Cardiac structure analysis from 3D micro-CT images”, while analyzing the micro-CT images of control avian heart vs the TOF avian heart, authors mention that in the TOF heart, 'right' branch of the pulmonary artery was missing compared to control heart. In support of this observation, they refer to a rare human condition 'unilateral absence of pulmonary artery' seen in TOF patients. However, in Figure 3 legend, they mention that the 'left' branch of pulmonary artery was absent in the TOF heart. Was this observation made in other samples? It is confusing to the reader and needs to be clarified.

We apologize for the typo. Observations were made from the same heart, and the left branch is missing. We have amended this mistake.

2) Article's main focus in on the multiscale imaging technique and its potential applications. The data presented in Figure 7 and Figure 8 are obtained from one control and one TOF avian heart. Although authors address this limitation in the article, one would need to be careful not to conclude much based on the observations comparing CT and TOF hearts although they are in line with previously published findings.

Thanks for pointing this important aspect of our work (Please note that Figure 7 and Figure 8 are now Figure 8 and Figure 9). We completely agree with the reviewer. Indeed, we cannot conclude anything of biological significance from just comparing one control and one TOF heart. We have performed some quantifications and analyses merely to show possibilities, but by no means suggesting we could make conclusions from them. We have now emphasized in the manuscript that the main focus is on the multiscale imaging, and the analysis performed on the imaging is to illustrate possibilities.

3) In subsection “Image segmentation and quantification”, authors mention while describing the 3D visualization of SBF-SEM segmentations (Figure 8), that orientation of myocardial cells were 'slightly' different. This needs to be described further as to how exactly the orientation of TOF myocardial cells was different from the control.

Thanks for pointing this out. The orientation of myocardial cells can be quantified, and because the ROIs are corresponding, these quantifications can be compared. We tried to do this visually in Figure 8 (now Figure 9) but we agree this is not enough. We have added to the explanation and have now included quantifications of myofibril orientation from both samples. These quantifications are summarized in Figure 10 but also described in the text (Results section and Materials and methods section).

4) Other potential applications of this novel imaging technique in addition to CHDs (extracellular matrix disorganization, fibrosis, myxomatous degeneration of valve tissues etc.) need to be briefly commented on. Any other cardiac ultra-structures that can potentially be studied using this technique must be mentioned briefly.

Thanks for making this important point, which we have not carefully discussed. Indeed, other applications of our imaging approach are possible, including the analysis of hearts that look normal but were subjected to insults early during development (are they completely normal?). As the reviewer points out, other applications of the multiscale imaging method are studies of extracellular matrix disorganization, fibrosis, and degeneration of valves. Moreover, in addition to studying nuclei, mitochondria and myofibrils (which we have emphasized in this manuscript as an example) we could also study the distribution (and amount) of lipid droplets, glycogen, as well as the ultrastructural organization within myocardial, endothelial, fibroblast and conduction cells.

We have added these points to the Discussion section.

“Outside the scope of this paper, but relevant to the comparison of normal versus CHD hearts, would be an extensive analysis of left and right ventricular wall microstructure and ultrastructure, including the distribution of lipid droplets, glycogen, and mitochondria with respect to the myofibrils. In addition, studies of the ultrastructural organization within myocardial, endocardial, fibroblast and conduction cells, in normal and CHD hearts, would be relevant to decipher the impact of CHD on cell and cardiac function.”

And the following to the Conclusions: “Importantly, multiscale studies can be used to decipher the imprints that early alterations in the environment in which the heart is growing have on cardiac formation and function. In addition, other potential applications to CHD and beyond are determinations of extracellular matrix organization/disorganization, cardiac fibrosis, glycogen distribution and myxomatous degeneration of valve tissues in response to different insults and aging.”

5) The novel imaging technique presented here improves on the previous methods by being applicable to larger tissues (up to 5-6mm according to authors). However, this is still smaller tissue size compared to other in vivo models and human heart tissues. Do authors foresee any potential problems in translating this technology to human heart tissues? This needs to be elaborated upon.

Thanks for this insightful comment. The method presented here is destructive, and as such (as presented) can only be applied to human samples of deceased individuals. Moreover, the larger size of the human heart will lead to difficulties in attaining homogeneous fixation and staining. To achieve fixation and stain homogeneity, increased timing for diffusion of fixative and heavy metal stains is certainly a possibility as is microwave steps (to accelerate diffusion). In addition, changes in processing, such as sectioning of hearts after microCT to facilitate diffusion of heavy metal stains and control imaging are also an option.

We have added the following paragraph to the Discussion section:

“Direct application of the proposed multiscale imaging method to human hearts is limited. The method presented here is destructive, and thus can only be applied to human samples of diseased individuals. Moreover, the larger size of the human heart will lead to difficulties in attaining homogeneous fixation and staining. To circumvent fixation and stain homogeneity issues, increased timing for diffusion of fixative and heavy metal stains is certainly a possibility as is microwave steps (to accelerate diffusion). In addition, changes in processing, such as sectioning of hearts after microCT to facilitate diffusion of heavy metal stains and control imaging are also possible. As presented, using our methods, multiscale imaging of human hearts is limited.”